# On Learning Fairness and Accuracy on Multiple Subgroups

**Changjian Shui**[1,2,4,*]   **Gezheng Xu**[3,*]  **Qi Chen**[4]   **Jiaqi Li**[3]
**Charles X. Ling**[3]   **Tal Arbel**[1,2,5]   **Boyu Wang**[3]   **Christian Gagné**[2,4,5]
[1]Centre for Intelligent Machines, McGill University       [2]Mila, Quebec AI Institute
[3]Department of Computer Science, University of Western Ontario
[4]Institute Intelligence and Data, Université Laval       [5]CIFAR AI Chair

## Abstract

We propose an analysis in fair learning that preserves the utility of the data while reducing prediction disparities under the criteria of group sufficiency. We focus on the scenario where the data contains multiple or even many subgroups, each with *limited number* of samples. As a result, we present a principled method for learning a fair predictor for all subgroups via formulating it as a bilevel objective. In the lower-level, the subgroup-specific predictors are learned through a small amount of data and the fair predictor. In the upper-level, the fair predictor is updated to be close to all subgroup specific predictors. We further prove that such a bilevel objective can effectively control the group sufficiency and generalization error. We evaluate the proposed framework on real-world datasets. Empirical evidence suggests the consistently improved fair predictions, as well as the comparable accuracy to the baselines.

## 1   Introduction

Machine learning has made rapid progress in sociotechnical systems such as automatic resume screening, video surveillance, and credit scoring for loan applications. Simultaneously, it has been observed that learning algorithms exhibited biased predictions on the *subgroups* of population [1, 2]. For example, the algorithm denies a loan application based on sensitive attributes such as gender, race, or disability, which has heightened public concerns.

To this end, fair learning is recently highlighted to mitigate prediction disparities. The high-level idea is quite straightforward: adding fair constraints during the training [3]. As a result, fair learning principally gives rise to two desiderata. On the one hand, the fair predictor should be *informative* to ensure accurate predictions for the data. On the other hand, the predictor is required to guarantee fairness to avoid prediction disparities across subgroups. Therefore, it is crucial to understand the possibilities and then design provable approaches for achieving both *informative* and *fair* learning.

Clearly, achieving both objectives depends on predefined fair notations. Consider demographic parity [1] as the fair criteria, which necessitates the independence between the predictor's output $f(X)$ and the sensitive attribute (or subgourp index) $A$. Thus, if the sensitive attribute $A$ and the ground-truth label $Y$ are highly correlated, it is impossible to learn a both fair and informative predictor.

To avoid such intrinsic impossibilities, alternative fair notions have been developed. In this work, we focus on the criteria of *group sufficiency* [1, 4], which ensures that the conditional expectation of ground-truth label ($\mathbb{E}[Y|f(X), A]$) is identical across different subgroups, given the predictor's output. Notably, the risk of violating group sufficiency has arisen in a number of real-world scenarios.

---

*Equal contribution

E.g., in medical artificial intelligence, the machine learning algorithm is used to assess the clinic risk, and guide decisions regarding initiating medical therapy. However, [5, 6] revealed a significant racial bias in such algorithms: when the algorithm predicts the same clinical risk score $f(X)$ for white and black patients, black patients are actually at a higher risk of severe illness: $\mathbb{E}[Y|f(X), A = \text{black}] \gg \mathbb{E}[Y|f(X), A = \text{white}]$. The deployed algorithms have resulted in more referrals of white patients to specialty healthcare services, resulting in both spending disparities and racial bias [5].

In summary, this work aims to propose a novel principled framework for ensuring group sufficiency, as well as preserving an informative prediction with a small generalization error. In particular, we focus on one challenge scenario: the *data includes multiple or even a large number of subgroups, some with only limited samples*, as often occurs in the real-world. For example, datasets for the self-driving car are collected from a wide range of geographical regions, each with a limited number of training samples [7]. How can we ensure group sufficiency as well as accurate predictions? Specifically, our contributions are summarized as follows:

**Controlling group sufficiency** We adopted *group sufficiency gap* to measure fairness w.r.t. group sufficiency of a classifier $f$ (Sec.3), and then derive an *upper bound* of the group sufficiency gap (Theorem 4.1). Under proper assumptions, the upper bound is controlled by the discrepancy between the classifier $f$ and the subgroup Bayes predictors. Namely, minimizing the upper bound also encourages an informative classifier.

**Algorithmic contribution** Motivated by the upper bound of the group sufficiency gap, we develop a principled algorithm. Concretely, we adopt a randomized algorithm that produces a predictive-distribution $Q$ over the classifier ($f \sim Q$) to learn informative and fair classification. We further formulate the problem as a bilevel optimization (Sec. 5.3), as shown in Fig.1. (1) In the lower-level, the subgroup specific dataset $S_a$ and the fair predictive-distribution $Q$ are used to learn the subgroup specific predictive-distribution $\overline{Q}_a^\star$, where $Q$ is regarded as an *informative prior* for learning limited data within each subgroup. Theorem 5.1 formally demonstrates that under proper assumptions, the lower-level loss can effectively control the generalization error. (2) In the upper-level, the fair predictive-distribution $Q$ is then updated to be close to all subgroup specific predictive-distributions, in order to minimize the upper bound of the group sufficiency gap.

Figure 1: Illustration of the proposed algorithm. Consider three subgroups $S_1, S_2, S_3$, e.g., datasets for three different races. The proposed algorithm is then formulated as a bilevel optimization to learn an informative and fair predictive-distribution $Q$. In the lower-level (cyan), we learn the subgroup specific predictive-distribution $\overline{Q}_a^\star$ from dataset $S_a$ (limited samples) and the prior $Q$. In the upper-level (brown), $Q$ is then updated to be as close to all of the learned subgroup specific $\overline{Q}_a^\star$ as possible.

**Empirical justifications** The proposed algorithm is applicable to the general parametric and differentiable model, where we adopt the neural network in the implementation. We evaluate the proposed algorithm on two real-world NLP datasets that have shown prediction disparities w.r.t. group sufficiency. Compared with baselines, the results indicate that group sufficiency has been consistently improved, with almost no loss of accuracy. Code is available at `https://github.com/xugezheng/FAMS`.

## 2 Related Work

**Algorithmic fairness** Fairness has been attached great importance and widely studied in various applications, such as natural language processing [8–10], natural language generation [11–13], computer vision [14, 15], and deep learning [16, 17]. Then various approaches have been proposed in algorithmic fairness. They typically add fair constraints during the training procedure, such as demographic parity or equalized odds [18–23]. Apart from this, other fair notions are adopted such as accuracy parity [24, 25], which requires each subgroup to attain the same accuracy; small prediction variance [26, 27], which ensures small prediction variations among the subgroup; or small prediction loss for all the subgroups [28–31]. Furthermore, based on the concept of Independence (e.g. demographic parity $A \perp\!\!\!\perp f(X)$) or conditional independence (e.g. equalized odds $A \perp\!\!\!\perp f(X)|Y$ or group sufficiency $A \perp\!\!\!\perp Y|f(X)$), another popular line in fair learning is then naturally integrated with information theoretical framework through adding mutual information constraints such as [32, 33].

**Understanding fairness-accuracy trade-off** As for the theoretical aspect, [34] further investigated the relation of fairness (demographic parity) and algorithmic stability. [35] formally justified the inherent trade-off between fairness (w.r.t. demographic parity and equalized odds) and accuracy, whereas the analysis is conducted for the binary sensitive attribute with the population loss. [36] studied the fair-accuracy trade-off in the multi-task learning.

**Group sufficiency** The fair notion of group sufficiency has recently been highlighted in various real-world scenarios such as health [6] and crime prediction [4, 37]. Specifically, [38] demonstrated that under proper assumptions, group sufficiency can be controlled in the unconstraint learning. However, this conclusion may not necessarily always hold in the overparameterized models with limited samples per subgroup, where [6, 39, 40] essentially revealed the prediction disparities between the different subgroups in the unconstraint learning. [41] recently studied the fair selective classification w.r.t. group sufficiency through an information theoretical framework, while the theoretical guarantee is unknown. In contrast, our proposed lower-level loss within the paper can provably control the generalization error, and the upper-level loss controls the group sufficiency gap. Besides, a close notion to the group sufficiency is the *probability calibration* [42], which is defined as $\mathbb{E}[Y|f(X)] = f(X)$ in binary classification. We will empirically show the probability calibration could be consistently improved within our framework, whereas the analysis on finite samples and its theoretical relation with group sufficiency remains still opening [43].

**Bi-level optimization in fairness** Bi-level optimization seeks to solve problems with a hierarchical structure. Namely, two levels of optimization problems where one task is nested inside another [44]. Several ideas related to bi-level optimization have been proposed in the context of fair-learning. For instance, we could design a min-max optimization to learn fair representation when considering demographic parity (DP) or equalized odds (EO) [19, 32, 25]. In this context, a representation function aims to minimize the loss caused by the discriminator in the lower-level. Simultaneously, in the upper-level, a discriminator could be introduced to maximize the loss. Then fair representation could be enforced through the bi-level optimization. Besides, if the accuracy and its variants are tracked as the metrics for each subgroup [12], the bi-level objective could also be deployed in controlling the loss [45] or the prediction variance [27], where the lower-level's goal is to minimize the loss for each subgroup and the upper-level's goal is to estimate the prediction disparities. In our paper, we *theoretically* justified a novel bi-level optimization perspective: controlling group sufficiency and accuracy. Simultaneously, other bi-level optimization and its relevant *meta-learning* algorithms could be further considered in the fair learning such as recurrent based gradient updating [46], layer-wise transformation [47] or implicit gradient based approach [48].

## 3 Preliminaries

We assume the joint random variable $(X, Y, A)$ follows an underlying distribution $\mathcal{D}(X, Y, A)$, where $X \in \mathcal{X}$ is the input, $Y \in \mathcal{Y}$ is the label, and the *scalar* discrete random variable $A \in \mathcal{A}$ denotes the sensitive attribute (or subgroup index). For instance, $A$ represents gender, race, or age. We also denote $\mathbb{E}[Y|X]$ as the conditional expectation of $Y$, which is essentially a function of $X$. $\mathbb{E}_{A,X}[\cdot]$ is denoted as the expectation on the marginal distribution of $\mathcal{D}(A, X)$. Throughout the paper, we consider binary classification with $\mathcal{Y} = \{0, 1\}$. We further define the predictor as a scoring function $f : \mathcal{X} \to [0, 1]$ that maps the input into a real value in $[0, 1]$. It is worth mentioning that in general $f(X) \notin \mathcal{Y}$ since $f(X)$ is continuous. We then introduce group sufficiency and group sufficiency gap.

**Definition 3.1** (Group sufficiency [1, 4, 38]). *A predictor $f$ satisfies group sufficiency with respect to the sensitive attribute $A$ if* $\mathbb{E}[Y|f(X)] = \mathbb{E}[Y|f(X), A]$.

Intuitively, given a output score of the predictor $f(X) = \tau$, the conditional expectation of $Y$ is invariant across different subgroups. Namely, conditioning on the specific subgroup $A = a$ does not provide any additional information about the conditional expectation of $Y$. Then we could naturally define group sufficiency gap.

**Definition 3.2** (Group sufficiency gap [38]). *The group sufficiency gap of a predictor $f$ is defined as:*
$$\mathbf{Suf}_f = \mathbb{E}_{A,X}[|\mathbb{E}[Y|f(X)] - \mathbb{E}[Y|f(X), A]|]$$

Specifically, $\mathbf{Suf}_f$ measures the extent of group sufficiency violation, induced by the predictor $f$, which is taken by the expectation over $(X, A)$. Clearly, $\mathbf{Suf}_f = 0$ suggests that $f$ satisfies groups sufficiency and vice versa. For completeness, we also discuss other popular group fairness criteria: demographic parity and equalized odds.

**Definition 3.3** (Demographic Parity (DP)). *A predictor $f$ satisfies the demographic parity with respect to the sensitive attribute $A$ if:* $\mathbb{E}[f(X)] = \mathbb{E}[f(X)|A]$

Demographic Parity (DP), also known as statistical parity or independence rule, emphasizes that the expectation of the output score $f(X)$ is independent of $A$. [1, 4] further revealed that if $A \not\perp Y$, group sufficiency and demographic parity could not be simultaneously achieved.

**Definition 3.4** (Equalized Odds (EO) [18]). *A predictor $f$ satisfies the equalized odds with respect to $A$ if:* $\mathbb{E}[f(X)|Y] = \mathbb{E}[f(X)|Y, A]$

Equalized odds (EO) emphasizes the conditional expectation of output $f$ is invariant w.r.t. $A$, given the ground truth $Y$. [1, 37] reveal that if $\mathcal{D}(X, Y, A) > 0$ and $A \not\perp Y$, group sufficiency and equalized odds can not both hold.

The analysis reveals a general *incompatibility* between group sufficiency and DP/EO when $A \not\perp Y$, which often occurs in practice. Besides, DP/EO based criteria generally suffers the well-known fair accuracy trade-off [32]: enforcing the fair constraint degrades the prediction performance. This paper depicts that under the criteria of group sufficiency, these objectives could be both encouraged.

## 4 Upper bound of group sufficiency gap

To derive the theoretical results, we first introduce the group Bayes predictor.

**Definition 4.1** ($A$-group Bayes predictor). *The $A$-group Bayes predictor $f_A^{Bayes}$ is defined as:* $f_A^{Bayes}(X) = \mathbb{E}[Y|X, A]$

The $A$-group Bayes predictor is associated with the underlying data distribution $\mathcal{D}(X, Y, A)$. Given the fixed realization $X = x, A = a$, we have $f_{A=a}^{\text{Bayes}}(x) = \mathbb{E}[Y|X = x, A = a]$, which suggests the ground truth conditional data generation of subgroup $A = a$. By adopting $f_{A=a}^{\text{Bayes}}(x)$, we could derive the upper bound of group sufficiency gap w.r.t. any predictor $f$:

**Theorem 4.1.** *Group sufficiency gap $\textbf{Suf}_f$ is upper bounded by:* $\textbf{Suf}_f \leq 4\mathbb{E}_{A,X}[|f - f_A^{Bayes}|]$
*Specifically, if $A$ takes finite value ($|\mathcal{A}| < +\infty$) and follows uniform distribution with $\mathcal{D}(A = a) = 1/|\mathcal{A}|$. Then the group sufficiency gap is further simplified as:*

$$\textbf{Suf}_f \leq \frac{4}{|\mathcal{A}|} \sum_a \mathbb{E}_X[|f - f_{A=a}^{Bayes}||A = a]$$

The proof is inspired by [38]. Specifically, Theorem 4.1 reveals that the upper bound of group sufficiency gap depends on the discrepancy between the predictor $f$ and $A$-group Bayes predictor $f_A^{\text{Bayes}}(X)$. Namely, given different subgroups $A = a$, the optimal predictor $f$ ought to be closed to all the group Bayes predictors $f_{A=a}^{\text{Bayes}}(X), \forall a \in \mathcal{A}$.

**Underlying assumption** Theorem 4.1 also reveals underlying assumptions w.r.t. the data generation distribution $\mathcal{D}(X, Y, A)$ for achieving a small group sufficiency gap. If $f_A^{\text{Bayes}}$ for each subgroup $A = a$ are quite similar, then minimizing the upper bound yields a small group sufficiency gap $\textbf{Suf}_f$. For example, consider the extreme scenario **if** the $A$-group Bayes predictors are identical w.r.t. $A$, $\mathbb{E}[Y|X, A = a] = \mathbb{E}[Y|X], \forall a \in \mathcal{A}$, where $\mathbb{E}[Y|X]$ is the conventional Bayes predictor defined on the marginalized distribution $\mathcal{D}(X, Y)$. The upper bound recovers the difference between the predictor $f$ and standard Bayes predictor. If we use a probabilistic framework to approximate predictor $f(X) \approx \mathbb{E}[Y|X]$ (i.e, training the entire dataset without any fair constraint), both group sufficiency gap and prediction error (since Bayes predictor is optimal) will be small, which is consistent with [38]. On the contrary, if $A$-group Bayes predictors are completely arbitrary with high variance for $A$, both group sufficiency gap and prediction error are large and it would be impossible for an informative prediction.

## 5 Principled Approach

Based on the upper bound, we propose a principled approach to learn the predictor that achieves both small generalization error and group sufficiency gap.

## 5.1 Upper bound in randomized algorithm

To establish the theoretical result, we consider a randomized algorithm that learns a predictive-distribution $Q$ over scoring predictors from the data. For instance, if we consider Bayes framework, the predictor is drawn from the posterior distribution $\tilde{f} \sim Q$. In the inference, the predictor's output is formulated as the expectation of the learned predictive-distribution $Q$: $f(X) = \mathbb{E}_{\tilde{f} \sim Q} \tilde{f}(X)$.

In practice, it is infeasible to optimize over all the possible distributions. Then we should restrict the predictive-distribution $Q$ within a distribution family $Q \in \mathcal{Q}$ such as Gaussian distribution. We also denote $Q_a^\star \in \mathcal{Q}$ as the optimal prediction-distribution w.r.t. $A = a$ under binary cross-entropy loss within the distribution family $\mathcal{Q}$: $Q_a^\star := \mathrm{argmin}_{Q_a \in \mathcal{Q}} \mathbb{E}_{\tilde{f}_a \sim Q_a} \mathcal{L}_a^{\mathrm{BCE}}(\tilde{f}_a)$. In generally $Q_a^\star \neq f^{\mathrm{Bayes}}(x, A = a)$, since the distribution family $\mathcal{Q}$ is only the subset of all possible distributions (shown in Fig. 2). We then extend the upper bound in the randomized algorithm.

**Corollary 5.1.** *The group sufficiency gap $\mathbf{Suf}_f$ in randomized algorithm w.r.t. learned predictive-distribution $Q$ is upper bounded by:*

$$\mathbf{Suf}_f \leq \frac{2\sqrt{2}}{|\mathcal{A}|} [\sum_a \underbrace{\sqrt{KL(Q_a^\star \| Q)}}_{Optimization} + \underbrace{\sqrt{KL(Q_a^\star \| \mathcal{D}(Y|X, A = a))}}_{Approximation}]$$

Where KL is the Kullback–Leibler divergence. Corollary 5.1 further reveals that the upper bound is decomposed into two terms, showing in Fig.2.

**Optimization term** The optimization term is the average KL divergence between the learned distribution $Q$ and optimal predictive-distribution $Q_a^\star$ for each subgroup $A = a$. Minimizing the optimization term implies that the learned distribution $Q$ will be both fair and informative for the prediction, because it aims to minimize the upper bound of the group sufficiency gap $\mathbf{Suf}_f$ and be close to the optimal predictive-distribution w.r.t. each $A = a$.

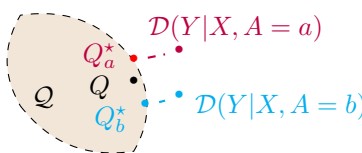

Figure 2: Illustration of optimization and approximation term. In the binary subgroup $\mathcal{A} = \{a, b\}$, the optimization term is to find $Q \in \mathcal{Q}$ to minimize the discrepancy between $(Q_a^\star, Q_b^\star)$. The approximation term is solely based on the distribution family $\mathcal{Q}$ (brown region). If the predefined $\mathcal{Q}$ has a rich expressive power, the approximation is treated as a small constant.

**Approximation term** The approximation term is the average KL divergence between the optimal distribution $Q_a^\star$ and the underlying data generation distribution. Given the distribution family $\mathcal{Q}$, it is a unknown constant. Besides, if the distribution family $\mathcal{Q}$ has a rich expressive power such as deep neural-network, the approximation term will be small [49]. However, an extreme large distribution family $\mathcal{Q}$ could simultaneously yield a potential overfitting on finite samples. In this paper, the neural network is adopted and the approximation term is assumed to be a small constant. Thus, controlling $\mathbf{Suf}_f$ implies minimizing the optimization term.

## 5.2 Challenge in learning limited samples

In practice, we only have access to finite or even limited samples in each subgroup, rather than the underlying distribution $\mathcal{D}$. We denote $S_a = \{(x_i^a, y_i^a)\}_{i=1}^m$ as the observed data w.r.t. subgroups $A = a$, which are i.i.d. samplings from the underlying distribution $\mathcal{D}(x, y | A = a)$. We also denote the **empirical** binary cross entropy loss w.r.t $A = a$ as: $\hat{\mathcal{L}}_a^{\mathrm{BCE}}(\tilde{f}) = \frac{1}{m} \sum_{i=1}^m -[y_i^a \log(\tilde{f}(x_i^a)) + (1 - y_i^a) \log(1 - \tilde{f}(x_i^a))]$. Then a straight approach is to minimize the empirical term $\hat{Q}_a^\star$:

$$\hat{Q}_a^\star = \mathrm{argmin}_{Q_a \in \mathcal{Q}} \mathbb{E}_{\tilde{f}_a \sim Q_a} \hat{\mathcal{L}}_a^{\mathrm{BCE}}(\tilde{f}_a) \tag{1}$$

Then $Q$ is updated through minimizing the average KL-divergence: $\sum_a \mathrm{KL}(\hat{Q}_a^\star \| Q)$ from learned $\hat{Q}_a^\star$. However, this idea generally *does not work* in our setting, because each subgroup contains **limited** number of samples. Therefore, a straight minimization leads to overfitting for each subgroup and generalization error $\mathbb{E}_{\tilde{f} \sim \hat{Q}_a^\star} \mathcal{L}_a^{\mathrm{BCE}}(\tilde{f})$ is quite large, showing in Fig. 5.

### 5.3 $Q$ as an informative prior

We have demonstrated that $Q$ can achieve both fair and informative prediction. Therefore, we regard $Q$ as a *prior* information for minimizing the loss, yielding a bilevel objective.

$$\min_{Q \in \mathcal{Q}} \frac{1}{|\mathcal{A}|} \sum_a \mathrm{KL}(\overline{Q}_a^{\star} \| Q) \qquad \text{(Upper-level)}$$

$$\text{s.t. } \overline{Q}_a^{\star} = \operatorname{argmin}_{Q_a \in \mathcal{Q}} \{ \mathbb{E}_{\tilde{f}_a \sim Q_a} \hat{\mathcal{L}}_a^{\mathrm{BCE}}(\tilde{f}_a) + \lambda \mathrm{KL}(Q_a \| Q) \}, \forall a \in \mathcal{A} \qquad \text{(Lower-level)}$$

Where $\lambda > 0$ is the hyper-parameter. The proposed loss is a typical bilevel optimization. (1) In the lower-level, we aim to learn $\overline{Q}_a^{\star}$ for each $a \in \mathcal{A}$. Different from Eq. (1), the loss in lower-level adds a regularization term $\mathrm{KL}(Q_a \| Q)$ as an informative prior in learning $\overline{Q}_a^{\star}$, given a fixed predictive-distribution $Q$. Moreover, Theorem 5.1 formally justified that optimizing the lower-level loss is to minimize the upper bound of the generalization error. (2) In the upper-level, $Q$ is updated through minimizing the average KL divergence between different $\overline{Q}_a^{\star}$, which controls the upper bound of $\mathbf{Suf}_f$.

**Theorem 5.1** (Generalization error bound)**.** *Supposing that datasets $\{S_a\}_{a=1}^{|\mathcal{A}|}$ with $S_a = \{(x_i^a, y_i^a)\}_{i=1}^m$ are i.i.d. sampled from $\mathcal{D}(x, y | A = a)$, the binary cross entropy (BCE) loss is upper bounded by $L$, $Q_a \in \mathcal{Q}$ is any learned distribution from dataset $S_a$ and $Q \in \mathcal{Q}$ is any distribution. Then with high probability $\geq 1 - \delta$ with $\forall \delta \in (0, 1)$, we have:*

$$\frac{1}{|\mathcal{A}|} \sum_a \mathbb{E}_{\tilde{f}_a \sim Q_a} \mathcal{L}_a^{BCE}(\tilde{f}_a) \leq \underbrace{\frac{1}{|\mathcal{A}|} \sum_a \mathbb{E}_{\tilde{f}_a \sim Q_a} \hat{\mathcal{L}}_a^{BCE}(\tilde{f}_a)}_{(1)} + \underbrace{\frac{L}{\sqrt{|\mathcal{A}|m}} \sum_a \sqrt{KL(Q_a \| Q)}}_{(2)} + \underbrace{L \sqrt{\frac{\log(1/\delta)}{|\mathcal{A}|m}}}_{(3)}$$

**Discussions** The proof is inspired by PAC-Bayes theorem such as [50–52]. Secpfically, Theorem 5.1 reveals the generalization error in the lower-level is upper bounded by three terms. (a) Term (1) is the average empirical prediction error, which corresponds to the first term in the lower-level loss. (b) Term (2) indicates the average KL-divergence between the learned subgroup distribution $Q_a$ and the prior distribution $Q$, which corresponds to the second term in the lower-level loss. The combination of term (1-2) recovers the averaged lower-level loss w.r.t. $A$. [2] Thus optimizing the lower-level loss could control the generalization error. (c) When the confidence $\delta$ is fixed, term (3) will converge if $|\mathcal{A}|m \to +\infty$. Moreover, even if $m$ (the sample size in each subgroup) is quite small, a sufficient large number of subgroups $|\mathcal{A}|$ can also ensure the convergence of term (3).

For the sake of simplicity, we assumed the identical samples size $m$ in each subgroup $S_a$, while the theoretical result can be extended to subgroups with different samples $m_a$.

### 5.4 Practical Implementations

In this section, we develop a practical learning algorithm that can be applied to a wide range of differentiable and parametric models, including neural networks.

**Parametric models** We choose the Isotropic Gaussian distribution (with diagonal covariance matrix) as the distribution family $\mathcal{Q}$, where the mean and covariance are set as $d$-dimensional parameter. Thus we need to learn the parameter $(\boldsymbol{\theta}, \boldsymbol{\sigma})$ for fair and informative $Q \in \mathcal{Q}$. As for the subgroup $A = a$, we learn parameters $(\boldsymbol{\theta}_a, \boldsymbol{\sigma}_a)$ for $\overline{Q}_a^{\star} \in \mathcal{Q}$. It is worth mentioning that the Isotropic Gaussian distribution is selected for its computational efficiency in the optimization. We can use any distribution as long as the density function is differentiable with respect to the parameters.

For the single predictor $\tilde{f}$, we use parametric neural-network models and assume $\tilde{f}$ is parameterized by a $d$-dimensional vector $\mathbf{w} \in \mathbb{R}^d$, denoted as $\tilde{f}_{\mathbf{w}}$. Then $\tilde{f}_{\mathbf{w}} \sim Q$ is equivalent to sampling the model parameter $\mathbf{w}$ from the predictive-distribution $Q$: $\mathbf{w} \sim \mathcal{N}(\boldsymbol{\theta}, \boldsymbol{\sigma}^2) = \prod_{i=1}^d \mathcal{N}(\boldsymbol{\theta}[i], \boldsymbol{\sigma}^2[i])$. Since $Q$ is Isotropic Gaussian, each element $i$ in the parameter $\mathbf{w}[i]$ follows a 1-dimensional Gaussian. Following the same line, $\tilde{f}_{\mathbf{w}_a} \sim \overline{Q}_a^{\star}$ can be modeled analogously: $\mathbf{w}_a \sim \mathcal{N}(\boldsymbol{\theta}_a, \boldsymbol{\sigma}_a^2) = \prod_{i=1}^d \mathcal{N}(\boldsymbol{\theta}_a[i], \boldsymbol{\sigma}_a^2[i])$.

---

[2]In Theorem 5.1, the differences are in the square norm of KL divergence and setting the specific hyper-parameter: $\lambda = L\sqrt{|\mathcal{A}|/m}$.

---

**Algorithm 1** Fair and Informative Learning for Multiple Subgroups (FAMS)

---

1: **Input:** Parameters w.r.t. distribution $Q$:$(\boldsymbol{\theta}, \boldsymbol{\sigma}^2)$, datasets $\{S_a\}$, $a \in \mathcal{A}$.
2: **for** Sampling a subset of $\{S_a\}$, where $a \in \mathcal{A}' \subseteq \mathcal{A}$ **do**
3:    ### Solving the lower-level ###
4:    Fix $Q$, optimizing the loss w.r.t. $Q_a = \mathcal{N}(\boldsymbol{\theta}_a, \boldsymbol{\sigma}_a^2)$ through SGD for each $a \in \mathcal{A}'$
$$\mathbb{E}_{\tilde{f}_{\mathbf{w}_a} \sim Q_a} \hat{\mathcal{L}}_a^{\text{BCE}}(\tilde{f}_{\mathbf{w}_a}) + \lambda \text{KL}(Q_a \| Q)$$
5:    Obtaining the solution $\overline{Q}_a^\star$, $a \in \mathcal{A}'$.
6:    ### Solving the upper-level ###
7:    Fix $\overline{Q}_a^\star$ with $a \in \mathcal{A}'$, optimizing the loss w.r.t. $Q$ through SGD: $\frac{1}{|\mathcal{A}'|} \sum_a \text{KL}(\overline{Q}_a^\star \| Q)$
8:    Obtaining updated parameter $(\boldsymbol{\theta}, \boldsymbol{\sigma}^2)$ in $Q$
9: **end for**
10: **Return:** Parameter of distribution $Q$: $(\boldsymbol{\theta}, \boldsymbol{\sigma}^2)$

---

As a result, learning the distribution $Q$ is equivalent to learning parameter $(\boldsymbol{\theta}, \boldsymbol{\sigma})$ in the bilevel objective.

**Gradient Estimation**   Based on the previous setting, we aim to optimize the bilevel objective to obtain the parameter of $Q$: $(\boldsymbol{\theta}, \boldsymbol{\sigma})$. We use stochastic gradient descent (SGD) to optimize the parameters. In the lower-level, the loss in Sec. 5.3 is composed by the empirical prediction error and KL divergence term. The KL divergence has a closed form that can be differentiated efficiently. Specifically, since $Q$ and the subgroup specific $\overline{Q}_a^\star$ are factorized Gaussian, the KL divergence takes a simple closed form and the gradient can be easily calculated: $\text{KL}(\overline{Q}_a^\star \| Q) = \frac{1}{2} \sum_{i=1}^d \left\{ \log \frac{\boldsymbol{\sigma}_a^2[i]}{\boldsymbol{\sigma}^2[i]} + \frac{\boldsymbol{\sigma}_a^2[i] + (\boldsymbol{\theta}_a[i] - \boldsymbol{\theta}[i])^2}{\boldsymbol{\sigma}^2[i]} - 1 \right\}$.

**Re-parametrization trick**   As for the prediction error $\mathbb{E}_{\tilde{f}_{\mathbf{w}_a} \sim Q_a} \hat{\mathcal{L}}_a^{\text{BCE}}(\tilde{f}_{\mathbf{w}_a})$, the term $\hat{\mathcal{L}}_a^{\text{BCE}}(\tilde{f}_{\mathbf{w}_a})$ is non-linear for $\mathbf{w}$, rendering the expectation intractable in the computation. To this end, we adopt the re-parameterization trick [53, 54] in computing the gradient w.r.t. the expectation term. The trick is based on describing the Gaussian distribution $\mathbf{w}_a \sim \mathcal{N}(\boldsymbol{\theta}_a, \boldsymbol{\sigma}_a^2)$ as first drawing $\boldsymbol{\epsilon} \sim \mathcal{N}(\mathbf{0}, \mathbf{I})$ and then applying the deterministic function $\mathbf{w}_a(\boldsymbol{\theta}_a, \boldsymbol{\sigma}_a) = \boldsymbol{\theta}_a + \boldsymbol{\sigma}_a \odot \boldsymbol{\epsilon}$ ($\odot$ is element-wise product) to approximate the sampling. Then the gradient term can be estimated as: $\nabla_{(\boldsymbol{\theta}_a, \boldsymbol{\sigma}_a)} \mathbb{E}_{\mathbf{w}_a \sim N(\boldsymbol{\theta}_a, \boldsymbol{\sigma}_a)} \hat{\mathcal{L}}_a^{\text{BCE}}(\tilde{f}_{\mathbf{w}_a}) = \nabla_{(\boldsymbol{\theta}_a, \boldsymbol{\sigma}_a)} \mathbb{E}_{\boldsymbol{\epsilon} \sim \mathcal{N}(\mathbf{0}, \mathbf{I})} \hat{\mathcal{L}}_a^{\text{BCE}}(\tilde{f}_{\mathbf{w}_a(\boldsymbol{\theta}_a, \boldsymbol{\sigma}_a)})$, where the expectation can be approximated by Monte-Carlo sampling w.r.t. $\boldsymbol{\epsilon}$. For a fixed sample $\boldsymbol{\epsilon}$, the gradient can be computed through backpropagation. In the upper-level, the KL divergence has a closed form, thus it is easy to update the parameter of $Q$ through backpropagation.

**Proposed Algorithm**   Based on the analysis, the algorithm is shown in Algorithm. 1 for solving the bilevel objective in Sec. 5.3. Specifically, we adopt the alternating optimization. Namely, in the lower-level, we fix $Q$ and optimize the subgroup specific predictor $\overline{Q}_a^\star$ through SGD. Then in the upper-level, we fix the learned $\overline{Q}_a^\star$ and update $Q$. Since we may face many subgroups, at each training epoch, we randomly sample a subset $\mathcal{A}'$ such that $|\mathcal{A}'| \ll |\mathcal{A}|$ for the memory saving.

**Inference**   In the inference, we use the Monte-Carlo method to sample the weights of the neural network from distribution $\mathbf{w} \sim \mathcal{N}(\boldsymbol{\theta}, \boldsymbol{\sigma}^2)$, then averaging the output w.r.t. different sampled weights to approximate $f(x) = \mathbb{E}_{\tilde{f}_{\mathbf{w}} \sim Q} \tilde{f}_{\mathbf{w}}(x)$

## 6 Experiments

### 6.1 Experimental Setup

**Dataset: Amazon review**   We adopt Amazon review dataset [55, 40], which aims to predict the sentiment (classification) from the review. The datasets consist of large-scale users. Each user has limited number of reviews, ranging from 75 to 400. The *user* is then treated as a subgroup, and it has been observed that standard training can lead to prediction disparities in several users.[56].The

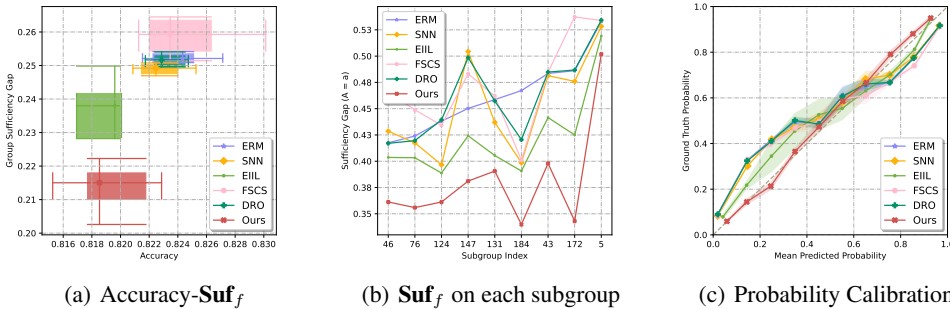

| (a) Accuracy-$\mathbf{Suf}_f$ | (b) $\mathbf{Suf}_f$ on each subgroup | (c) Probability Calibration |

Figure 3: Amazon Review dataset. (a) Boxplot of accuracy and group sufficiency gap $\mathbf{Suf}_f$ with 5 repeats: median, 75th percentile and minimum-maximum value. (b) Group sufficiency gap on subgroup $A = a$, which is the difference between $\mathbb{E}[Y|f(X)]$ and $\mathbb{E}[Y|f(X), A = a]$. We visualize the top-9 users' group sufficiency gap in ERM, whereas the result for all users is delegated to the Appendix. (c) Probability calibration curve over 5 repeats with mean and standard deviation. i.e $(f(X), \mathbb{E}[Y|f(X)])$. The proposed approach demonstrated a consistently improved probability calibration.

experiment is adapted from the protocol of [40]. Specifically, we convert the original review score (ranging from 1-5) to the binary label: the positive review (score $\geq 4$) and negative review (score $\leq 3$). We draw and then fix 200 users from the original dataset, which includes the training, validation, and test sets. In the implementation, we first adopt DistilBERT [57] to learn the embedding with dimension $\mathbb{R}^{768}$. Then we adopt $\tilde{f}_{\mathbf{w}}$ and $\tilde{f}_{\mathbf{w_a}}$ as the four-layer fully connected neural network, where $\mathbf{w} \sim Q$ and $\mathbf{w}_a \sim Q_a$. Additional experimental details are delegated to the Appendix.

**Dataset: Toxic Comments**  We also use the toxic comment dataset [58] to predict the text comment being toxic or not, which has shown the significant performance degradation on specific sub-populations. Following [58], we select *race* as sensitive attribute, which includes Black, White, Asian and Latino & others (4 subgroups). We also follow the same setting as the original dataset [40], which has the separate training, validation, and test sets. Since toxic comments are marked by multiple annotators, we decide that the comment is toxic if it is marked by at least half of the annotators. In the implementation, we adopt the DistilBERT [57] as the embedding with output dimension $\mathbb{R}^{768}$. Then we also adopt $\tilde{f}_{\mathbf{w}}$ and $\tilde{f}_{\mathbf{w_a}}$ as the four-layer fully connected neural network, with the same network structure as Amazon. Additional details are delegated to the Appendix.

**Baselines**  We compare with following baselines. (1) ERM: training a deep model without considering the sensitive attribute. (2) SNN. Since we adopt randomized algorithms in the paper, we additionally compare the stochastic neural network through the vanilla training from the whole dataset. Namely, we find a predictive-distribution $Q$ to minimize $\frac{1}{|\mathcal{A}|} \sum_a \mathbb{E}_{f \sim Q} \hat{\mathcal{L}}_a^{\text{BCE}}(f)$. (3) EIIL [43] proposed an IRM based approach to promote the group sufficiency. (4) FSCS [41] adopted the conditional mutual information constraint $I(A, Y|f(X))$ to promote the sufficiency. (5) DRO [24]. A re-weighting approach to assign the importance of the task. Indeed, DRO does not provably guarantee group sufficiency, while it encourages identical losses. All the experiments are repeated five times.

**Computing $\mathbf{Suf}_f$**  Since $f(X)$ is continuous, the group sufficiency gap is calculated by splitting the output of predictor into multiple intervals in $[0, 1]$ and computing the conditional expectation within each interval, as detailed in Appendix.

## 6.2 Experimental Results

We visualize the results in Fig. 3 for Amazon review product and Fig. 4 for toxic comment.

**Accuracy and Fairness** The accuracy and group sufficiency gap are depicted in Fig. 3(a) and Fig. 4(a). In Amazon review, the accuracy in the proposed approach has a slight decrease, compared with ERM.

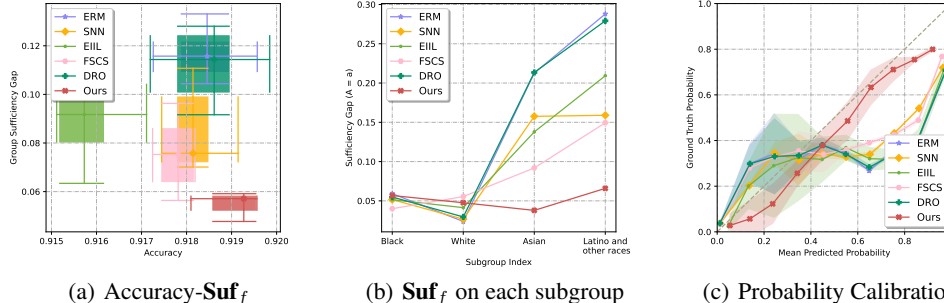

(a) Accuracy-**Suf**$_f$      (b) **Suf**$_f$ on each subgroup      (c) Probability Calibration

Figure 4: Toxic dataset. (a) Boxplot of accuracy and group sufficiency gap **Suf**$_f$ with 5 repeats: median, 75th percentile and minimum-maximum value. (b) Group sufficiency gap on the specific subgroup $A = a$, which is the difference between $\mathbb{E}[Y|f(X)]$ and $\mathbb{E}[Y|f(X), A = a]$. (c) Probability calibration over 5 repeats with mean and standard deviation. i.e $(f(X), \mathbb{E}[Y|f(X)])$, where the probability calibration for each subgroup is delegated to Appendix.

While the group sufficiency gap has improved by $3.0\%$, showing a significant improvement in the fairness. In toxic comments, the accuracy in proposed approach is nearly identical to the baseline, whereas group sufficiency gap has been significantly improved by $3.0\text{-}3.5\%$.

**Group sufficiency gap on the specific subgroup** To gain better understandings of group sufficiency gap, we visualize group sufficiency gap on specific subgroup $A = a$, i.e the discrepancy between the $\mathbb{E}[Y|f(X)]$ (conditional expectation on the entire data) and $\mathbb{E}[Y|f(X), A = a]$ (conditional expectation on subgroup $A = a$). I.e, $\mathbb{E}_X\left[|\mathbb{E}[Y|f(X)] - \mathbb{E}[Y|f(X), A = a]|\right]$.

In Amazon review dataset, we visualize the top-9 users' sufficiency gap in ERM, as shown in Fig. 3(b), where the gap of entire users is delegated to the Appendix. The proposed approach significantly reduces the group sufficiency gap of in most subgroups. The similar trend is also observed in Toxic dataset, as shown in Fig. 4(b), where the proposed approach has the nearly identical and small group sufficiency gap for each race. In contrast, the baselines exhibit significant group sufficiency bias on the Asian and Latino & other races.

**Probability Calibration** A related concept to group sufficiency is the probability calibration [42], which is defined as $\mathbb{E}[Y|f(X)] = f(X)$ in the binary classification. We visualize the probability sufficiency of Amazon review in Fig. 3(c) and Toxic comments in Fig. 4(c). The results suggest that the proposed approach demonstrates a consistently better probability calibration on the whole data. We then visualize the probability calibration for each subgroup, as shown in the Appendix, where the results reveal the improved probability calibration for each subgroup.

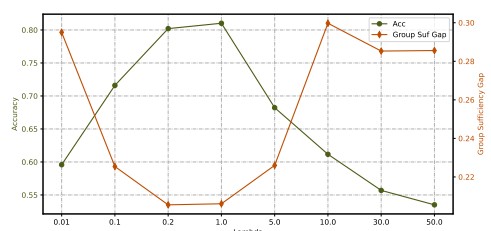

Figure 5: Analysis. Accuracy-**Suf**$_f$ curve under different $\lambda$ in Amazon review dataset.

**Other sensitive attributes in Toxic comments** Apart from adopting race as sensitive attribute, we also consider other possible sensitive attributes such as gender and religion, and the results are showed in the Appendix. The results in other sensitive attributes are similar to race, with improved fairness and no loss on accuracy.

**Influence of $\lambda$. Fairness and accuracy can be simultaneously achieved.** Theorem 5.1 suggests that there exists an optimal $\lambda$ in the generalization error bound. Then we changed the value of $\lambda$ in Amazon dataset, as shown in Fig. 5.

When $\lambda \to 0$, the subgroup specific parameters are simply learned from the limited samples within each subgroup. Then the fair predictor could not learn a proper prior from the subgroup specific predictor with a significant generalization error. Meanwhile, the group sufficiency gap is also large,

which is consistent with [38]: overfitting generally degrades the group sufficiency. When $\lambda$ is set between $[0.2, 1]$, the generalization error is small (with a high accuracy) and group sufficiency gap is kept small, implying that both fairness and accuracy can be achieved. In contrast, if we set a large value for $\lambda \gg 0$, the predictor is unable to learn from the data but from the random prior $Q$. The prediction will be completely random (accuracy $= 55\%$ when $\lambda = 50$). When the predictor outputs a random guess, different from demographic parity (DP) or equalized odds (EO), the group sufficiency gap is also large. The analysis reveals that there exists an optimal $\lambda$ for simultaneously achieving accuracy and group sufficiency.

## 7    Conclusion

We conducted a novel analysis by simultaneously learning an informative and fair classifier for multiple or even many subgroups. We derived a novel principled algorithm. We further theoretically justified the generalization error and fair guarantees of the proposed framework. The empirical results in two real-world datasets demonstrated the effectiveness in both preserving the accuracy, as well as group sufficiency.

### Discussion on Limitations

We proposed the analysis on learning group sufficiency and informative predictors, and developed a principled approach for it. Simultaneously there are several limitations to the proposed theory and algorithm. (1) In general, group sufficiency and DP/EO are incompatible. Controlling group sufficiency, for example, would cause DP/EO degradation. This would be problematic if DP/EO were preferred in practice. (2) We also assumed that the ground truth $A$-Bayes predictors would be similar across groups. However, this assumption could be violated, resulting in a highly non-trivial scenario. Thus, in order to evaluate the conditional distribution shift, we need to consider a new setting by collect sufficient data per subgroup.

### Acknowledgments and Disclosure of Financial Support

We appreciate constructive feedback from anonymous reviewers and meta-reviewers. We also would like to thank Jun Xiao for the discussion and proof-reading the manuscript. C. Shui and C. Gagné acknowledge support from NSERC-Canada and the Canada CIFAR Chairs in AI. G. Xu, J. Li, C. Ling and B. Wang are supported by Natural Sciences and Engineering Research Council of Canada (NSERC), Discovery Grants program. T. Arbel is supported by International Progressive Multiple Sclerosis Alliance, the Canada Institute for Advanced Research (CIFAR) Artificial Intelligence Chairs program, the Natural Sciences and Engineering Research Council of Canada. Q. Chen is supported by China Scholarship Council.

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
