# A   Group sufficiency vs. demographic parity (DP) and equalized odds (EO)

For better understanding the properties of these three metrics, we consider the following scenario.

Consider the score predictor $f(X)$ uniformly outputs a value in $[0,1]$ for any $x \in \mathcal{X}$. i.e, $f(x) \sim$ Unif$[0,1]$. Then it is easy to verify the demographic party and equalized odds both satisfy. Since the output of predictor is completely independent with the data. Thus we have

$$\mathbb{E}[f(X)] = \mathbb{E}[f(X)|A], \mathbb{E}[f(X)|Y] = \mathbb{E}[f(X)|A,Y]$$

In contrast, the group sufficiency does not necessarily hold. Specifically we have $\mathbb{E}[Y|f(X)] = \mathbb{E}[Y], \mathbb{E}[Y|f(X), A] = \mathbb{E}[Y|A]$, then we have

$$\mathbf{Suf}_f = \mathbb{E}_A|\mathbb{E}[Y] - \mathbb{E}[Y|A]| > 0,$$

if $A \not\perp Y$.

The aforementioned counterexample suggests that group sufficiency shows different behaviors, where the pure random prediction could trivially achieve the EO/DP.

Moreover, based on [1], if $\mathcal{D}(X, Y, A) > 0$ and $A \not\perp Y$, the group sufficiency and demographic parity/equalized odds do not both hold. The aforementioned example also verifies this fact.

# B   Additional Facts of $A$-Group Bayes predictor

For better understanding $A$-Group Bayes predictor, we could derive the following facts.

**Proposition B.1.** *The $A$-group Bayes predictor $f_A^{Bayes}(X)$: (1) Given a subgroup $A = a$, satisfies the group sufficiency $\mathbb{E}[Y|f_{A=a}^{Bayes}(X)] = \mathbb{E}[Y|f_{A=a}^{Bayes}(X), A = a]$; (2) is the optimal predictor under subgroup $A = a$ the binary cross-entropy loss function.* [3] *Namely, $f_{A=a}^{Bayes}(x) = argmin_f \mathcal{L}_a^{BCE}(f)$, where $\mathcal{L}_a^{BCE}(f) = \mathbb{E}_{(x,y) \sim \mathcal{D}(x,y|A=a)} - [y\log(f(x)) + (1-y)\log(1-f(x))]$.*

Proposition B.1 shows that given a subgroup $A = a$, $A$-group Bayes predictor simultaneously meets both fairness (group sufficiency) and informative (optimal predictor under binary cross-entropy loss). Unfortunately, $f^{\text{Bayes}}$ is impossible to estimate since it is related to the *underlying data distribution* $\mathcal{D}$, which is infeasible. Nevertheless, we can adopt $A$-group Bayes predictor $f^{\text{Bayes}}$ to derive an upper bound of group sufficiency gap $\mathbf{Suf}_f$. The upper bound then holds for any predictor $f$ that can be learned from the observed data.

## B.1   Proof of Fact 1

We first introduce the generalized Tower rule of the conditional expectation.

**Generalized tower rule of conditional expectation**   Let $(\Omega, \mathcal{F}, P)$ be the probability space and two sub $\sigma$-algebras $\mathcal{G}_1 \subseteq \mathcal{G}_2 \subseteq \mathcal{F}$ are defined. Then we have

$$\mathbb{E}[\mathbb{E}[X|\mathcal{G}_2]|\mathcal{G}_1] = \mathbb{E}[X|\mathcal{G}_1]$$

*Proof.* Based on the generalized tower rule, we have

$$\mathbb{E}[Y|f_A^{\text{Bayes}}(X), A] = \mathbb{E}[\mathbb{E}[Y|X, A, f_A^{\text{Bayes}}(X)]|f_A^{\text{Bayes}}(X), A]$$
$$= \mathbb{E}[\mathbb{E}[Y|X, A]|f_A^{\text{Bayes}}(X), A] = \mathbb{E}[f_A^{\text{Bayes}}|f_A^{\text{Bayes}}(X), A] = f_A^{\text{Bayes}}(X)$$

Also we have:

$$\mathbb{E}[Y|f_A^{\text{Bayes}}(X)] = \mathbb{E}[\mathbb{E}[Y|f_A^{\text{Bayes}}(X), A, X]|f_A^{\text{Bayes}}(X)]$$
$$= \mathbb{E}[\mathbb{E}[Y|f_A^{\text{Bayes}}(X), A, X]|f_A^{\text{Bayes}}(X)]$$
$$= \mathbb{E}[\mathbb{E}[Y|A, X]|f_A^{\text{Bayes}}(X)] = \mathbb{E}[f_A^{\text{Bayes}}(X)|f_A^{\text{Bayes}}(X)] = f_A^{\text{Bayes}}(X)$$

---

[3]The binary cross-entropy is chosen as the prediction loss because it is widely used in classification. In fact, $f_A^{\text{Bayes}}$ is also the optimal predictor under square loss for each subgroup $A = a$. The entire analysis can be extended to the regression with square loss.

Combining these two equations, we have the Fact 1:

$$\mathbb{E}[Y|f_A^{\text{Bayes}}(X), A] = \mathbb{E}[Y|f_A^{\text{Bayes}}(X)]$$

The aforementioned proof adopts the fact $\mathbb{E}[Y|f_A^{\text{Bayes}}(X), A, X] = \mathbb{E}[Y|A, X]$, since the conditional expectation is uninfluenced by the $A$-group Bayes predictor, given $A, X$. □

## B.2 Proof of Fact 2

Following [59], we can compute the optimal predictor of attribute $A = a$ under the binary cross-entropy by taking the functional derivative w.r.t. $f$:

$$\frac{d\mathcal{L}_a^{\text{BCE}}(f)}{df} = 0$$

We have

$$\mathbb{E}_{\mathcal{D}(x,y|a)}[f(x) - y] = 0$$
$$\rightarrow \mathbb{E}_{\mathcal{D}(x|a)}[f(x)\mathbb{E}_{\mathcal{D}(y|x,a)} - \mathbb{E}_{\mathcal{D}(y|x,a)}y] = 0$$
$$\rightarrow \mathbb{E}_{\mathcal{D}(x|a)}[f(x) - \mathbb{E}_{\mathcal{D}(y|x,a)}y] = 0$$

Therefore, we have the optimal predictor $f^\star = \mathbb{E}_{\mathcal{D}(y|x,a)}y = \mathbb{E}[Y = y|X = a, A = a]$, the $A$-group Bayes predictor. [59] further demonstrated the optimal is unique under the binary cross entropy loss.

## C Upper bound of group sufficiency gap

Before deriving the theory, we need the following lemma.

**Lemma C.1.** *For any predictor $f$, we have*

$$\mathbb{E}[Y|f(X), A] = \mathbb{E}[f_A^{Bayes}(X)|f(X), A]$$

*Proof.*

$$\begin{aligned}
\mathbb{E}[Y|f(X), A] &= \mathbb{E}[\mathbb{E}[Y|f(X), A, X]|f(X), A] \\
&= \mathbb{E}[\mathbb{E}[Y|A, X]|f(X), A] \\
&= \mathbb{E}[f_A^{\text{Bayes}}(X)|f(X), A]
\end{aligned}$$

□

Based on Lemma B.1, we can derive the main Theorem.

*Proof.* For the simplicity, we denote $\mathbb{E}[Y|f, A] = \mathbb{E}[Y|f(X), A]$ and $\mathbb{E}[Y|f_A^{\text{Bayes}}, A] = \mathbb{E}[Y|f_A^{\text{Bayes}}(X), A]$. We first bound $\mathbb{E}_{A,X}[|\mathbb{E}[Y|f, A] - \mathbb{E}[Y|f_A^{\text{Bayes}}, A]|]$

$$\begin{aligned}
\mathbb{E}_{A,X}[|\mathbb{E}[Y|f, A] - \mathbb{E}[Y|f_A^{\text{Bayes}}, A]|] &= \mathbb{E}_{A,X}[|\mathbb{E}[f_A^{\text{Bayes}}|f, A] - f_A^{\text{Bayes}}|] \\
&= \mathbb{E}_{A,X}[|\mathbb{E}[f_A^{\text{Bayes}} - f|f, A] + f - f_A^{\text{Bayes}}|] \\
&\leq \mathbb{E}_{A,X}[|\mathbb{E}[f_A^{\text{Bayes}} - f|f, A]| + |f - f_A^{\text{Bayes}}|] \\
&= 2\mathbb{E}_{A,X}[|f - f_A^{\text{Bayes}}|]
\end{aligned}$$

Where $\mathbb{E}_{A,X}[|\mathbb{E}[f_A^{\text{Bayes}} - f|f, A]|] = \mathbb{E}_{A,X}[\mathbb{E}[f_A^{\text{Bayes}} - f|f, A, X]] = \mathbb{E}_{A,X}[|\mathbb{E}[f_A^{\text{Bayes}} - f|A, X]|] = \mathbb{E}_{A,X}[|f_A^{\text{Bayes}} - f|]$ is derived from the tower rule of conditional expectation.

Analogously, we can bound

$$\mathbb{E}_{A,X}[|\mathbb{E}[Y|f] - \mathbb{E}[Y|f_A^{\text{Bayes}}]|] \leq 2\mathbb{E}_{A,X}[|f - f_A^{\text{Bayes}}|]$$

Thus the group sufficiency gap is upper bounded by:

$$
\begin{aligned}
\mathbf{Suf}_f &= \mathbb{E}_{A,X}[|\mathbb{E}[Y|f] - \mathbb{E}[Y|f,A]|] \\
&= \mathbb{E}_{A,X}[|\mathbb{E}[Y|f] - \mathbb{E}[Y|f,A] - \mathbb{E}[Y|f_A^{\text{Bayes}}] - \mathbb{E}[Y|f_A^{\text{Bayes}},A]|] \quad \text{(Using Fact 1)} \\
&\leq \mathbb{E}_{A,X}[|\mathbb{E}[Y|f] - \mathbb{E}[Y|f_A^{\text{Bayes}}]| + |\mathbb{E}[Y|f,A] - \mathbb{E}[Y|f_A^{\text{Bayes}},A]|] \\
&\leq 4\mathbb{E}_{A,X}[|f - f_A^{\text{Bayes}}|]
\end{aligned}
$$

Therefore, if $A$ takes only finite value ($|\mathcal{A}| < +\infty$) and follows uniform distribution with $\mathcal{D}(A = a) = 1/|\mathcal{A}|$, then we have:

$$
\mathbf{Suf}_f \leq \frac{4}{|\mathcal{A}|} \sum_a \mathbb{E}_X[|f - f_{A=a}^{\text{Bayes}}| |A = a]
$$

$\square$

## D  Upper bound of group sufficiency gap in randomized algorithm

According to the definition, we have:

$$
\begin{aligned}
\mathbf{Suf}_f &\leq \frac{4}{|\mathcal{A}|} \sum_a \mathbb{E}_X[|\mathbb{E}_{\tilde{f}\sim Q}\tilde{f}(x) - \mathbb{E}_{\tilde{f}\sim Q_a^\star}\tilde{f}(x)| + |\mathbb{E}_{\tilde{f}\sim Q_a^\star}\tilde{f}(x) - \mathbb{E}_{\tilde{f}\sim \mathcal{D}(y|x,a)}\tilde{f}(x)|] \\
&\leq \frac{4}{|A|} \sum_a \text{TV}(Q_a^\star\|Q) + \text{TV}(Q_a^\star\|\mathcal{D}(y|x,a)) \quad \text{(Property of Total variation distance)} \\
&\leq \frac{2\sqrt{2}}{|A|}[\sum_a \underbrace{\sqrt{\text{KL}(Q_a^\star\|Q)}}_{\text{Estimation Error}} + \underbrace{\sqrt{\text{KL}(Q_a^\star\|\mathcal{D}(y|x,a))}}_{\text{Approximation Error}}] \quad \text{(Pinsker's inequality)}
\end{aligned}
$$

The second line is derived from the property of Total variation distance. Note the scoring predictor ranges in $[0, 1]$: $\tilde{f}(X = x) \in [0, 1]$.

The third line is derived from Pinsker's inequality. i.e, $\text{TV}(P\|Q) \leq \sqrt{\frac{1}{2}\text{KL}(P\|Q)}$.

## E  Generalization upper bound

**Step 1**  We first demonstrate the following Lemma, which is based on [50, 60].

**Lemma E.1.** *Let $f$ be a random variable taking value in $A$ and let $X_1, \ldots, X_l$ be $l$ independent variables with each $X_k$ distributed to $\mu_k$ over the set $A_k$. For function $g_k : A \times A_k \to [a_k, b_k]$, $k = 1, \ldots, l$. Let $\zeta_k(f) = \mathbb{E}_{X_k\sim\mu_k} g_k(f, X_k)$ for any fixed value of $f$. Then for any fixed distribution $\pi$ on $A$ and any $\lambda, \delta > 0$, the following inequality holds with high probability $1 - \delta$ over the sampling $X_1, \ldots, X_l$ for all distribution $\rho$ over $A$.*

$$
\mathbb{E}_{f\sim\rho} \sum_{k=1}^l \zeta_k(f) - \mathbb{E}_{f\sim\rho} \sum_{k=1}^l g_k(f, X_k) \leq \frac{1}{\lambda}\left(KL(\rho\|\pi) + \frac{\lambda^2}{8}\sum_{k=1}^l (b_k - a_k)^2 + \log\frac{1}{\delta}\right)
$$

**Step 2**  Then we could use the aforementioned Lemma to demonstrate the main theorem.

*Proof.* We adopt the lemma for the union of the whole training samples $S = \cup_{a\in\mathcal{A}}S_a$.

We set

$$
\rho = \underbrace{(Q_1 \otimes Q_2 \otimes \cdots \otimes Q_{|\mathcal{A}|})}_{|\mathcal{A}| \text{ times}} \qquad \pi = \underbrace{(Q \otimes Q \otimes \cdots \otimes Q)}_{|\mathcal{A}| \text{ times}}
$$

We also set $X_k = (x_i^a, y_i^a)$, $l = |\mathcal{A}|m$, $f = (\tilde{f}_1, \ldots, \tilde{f}_a, \ldots, \tilde{f}_{|\mathcal{A}|})$, $g_k(f, X_k) = \frac{1}{|\mathcal{A}|m}\ell^{\text{BCE}}(\tilde{f}_a(x_i^a), y_i^a)$. Since we adopt the binary cross entropy loss, $a_k = 0$ and $b_k = L/(|\mathcal{A}|m)$,

then with high probability $1 - \delta$, we have:

$$\frac{1}{|\mathcal{A}|}\sum_a \mathbb{E}_{\tilde{f}_a \sim Q_a} \mathcal{L}_a^{\mathrm{BCE}}(\tilde{f}_a) \leq \frac{1}{|\mathcal{A}|}\sum_a \mathbb{E}_{\tilde{f}_a \sim Q_a} \hat{\mathcal{L}}_a^{\mathrm{BCE}}(\tilde{f}_a)$$
$$+ \frac{1}{\lambda}(\mathrm{KL}(Q_1 \otimes \cdots \otimes Q_{|\mathcal{A}|} \| Q \otimes \cdots \otimes Q) + \log(\frac{1}{\delta})) + \frac{\lambda L}{8|\mathcal{A}|m}$$

Through the decomposition property of KL divergence, we finally have:

$$\frac{1}{|\mathcal{A}|}\sum_a \mathbb{E}_{\tilde{f} \sim Q_a} \mathcal{L}_a^{\mathrm{BCE}}(\tilde{f}) \leq \frac{1}{|\mathcal{A}|}\sum_a \mathbb{E}_{\tilde{f} \sim Q_a} \hat{\mathcal{L}}_a^{\mathrm{BCE}}(\tilde{f}) + L\sqrt{\frac{1}{2|\mathcal{A}|m}(\sum_a \mathrm{KL}(Q_a\|Q) + \log(\frac{1}{\delta}))}$$

$$\leq \frac{1}{|\mathcal{A}|}\sum_a \mathbb{E}_{\tilde{f} \sim Q_a} \hat{\mathcal{L}}_a^{\mathrm{BCE}}(\tilde{f}) + \frac{L}{\sqrt{|\mathcal{A}|m}}\sum_a \sqrt{\mathrm{KL}(Q_a\|Q)} + L\sqrt{\frac{\log(1/\delta)}{|\mathcal{A}|m}}$$

$$\square$$

# F   Computing group sufficiency gap from the data

In this paper, we need to compute the conditional expectation from the data. i.e,

$$\mathbb{E}[Y|f(X)], \quad \mathbb{E}[Y|f(X), A = a],$$

where we have observed data $\{S_a\}, a \in \mathcal{A}$. Since $f(x)$ is a continuous value, ranging from $[0, 1]$. Then we split $[0, 1]$ into sperate interevals:

$$[0, \epsilon_1], [\epsilon_1, \epsilon_2], \ldots, [\epsilon_N, 1]$$

We compute the expectation and conditional expectation within each interval. i.e:

$$(\mathbb{E}[f(X)\mathbf{1}_{\{f(X)\in[\epsilon_i,\epsilon_{i+1}]\}}], \mathbb{E}[Y|f(X) \in [\epsilon_i, \epsilon_{i+1}]]) = (p_i, q_i)$$

$$(\mathbb{E}[f(X)\mathbf{1}_{\{f(X)\in[\epsilon_i,\epsilon_{i+1}], A=a\}}], \mathbb{E}[Y|f(X) \in [\epsilon_i, \epsilon_{i+1}], A = a]) = (p_i^a, q_i^a)$$

Then for each group $A = a$, the group sufficiency gap is computed as:

$$\mathbf{Suf}_f(A = a) = \sum_i |q_i - \underbrace{(q_i^a + \frac{p_i - p_i^a}{p_{i+1}^a - p_i^a}(q_i^{a+1} - q_i^a))}_{\text{Linear Interpolation}}|$$

We use the linear interpolation if the average values in each interval are not equal. Then the group sufficiency can be formulated as:

$$\mathbf{Suf}_f = \frac{1}{|\mathcal{A}|}\sum_a \mathbf{Suf}_f(A = a)$$

We assign the $\mathcal{D}(A = a) = \frac{1}{|\mathcal{A}|}$ as uniform distribution for ensuring fairness for each subgroup.

**Discussions**   In general:

$$\mathbb{E}[Y|f(X)] \neq \frac{1}{|\mathcal{A}|}\sum_a \mathbb{E}[Y|f(X), A = a]$$

The demonstration is straightforward. By using the Bayes rule, we have

$$\mathbb{E}[Y|f(X)] = \sum_a \mathcal{D}(A = a|f(X))\mathbb{E}[Y|f(X), A = a]$$

Thus iff $\mathcal{D}(A = a|f(X)) = \frac{1}{|\mathcal{A}|}$, we have the equivalent form. Intuitively, $\mathcal{D}(A = a|f(X))$ refers the conditional probability of $A = a$, given the predicted score $f(X)$, which is related to the group membership inference such as [61]. If $\mathcal{D}(A = a|f(X))$ is large, the subgroup index can be easily revealed via the algorithm output. If the algorithm can fully preserve the privacy, then $\mathcal{D}(A = a|f(X)) = \frac{1}{|\mathcal{A}|}$.

# G Experimental Details

In this part, we proposed a detailed description of the dataset and experiments settings.

## G.1 Amazon review dataset

The experiment is adapted from the protocol of [40]. Specifically, we convert the original review score (ranging from 1-5) to the binary label: the positive review (score $\geq 4$) and negative review (score $\leq 3$). We sample and then fix 200 users from the original dataset, which contains the training (75-400 samples per user), validation (75 samples per user), and test sets (75 samples per user).

The total training epoch is 100. In each training epoch, we sample a small subset of users ($N_{\text{user}} = 20$), then for each user we sample 50 samples with replacement. The early stopping strategy is also adopted.

We adopt 4-layers fully connected neural network as the model, where the weights of the model follows the Gaussian distribution $Q_a$ or $Q$. The trade-off coefficient $\lambda$ ranges from $[0.01, 50]$ and we fix $\lambda = 0.4$ in the evaluation. We set all the Monte-Carlo samples as 5. More implementation details of experiments and parameter settings can be found in the code.

## G.2 Toxic comments

We adopt the toxic comment dataset [58] to predict whether the text comment is toxic or not, which has been observed the significant performance degradation on particular sub-populations.

Following [58, 40], we first choose the *race* as sensitive attribute, which includes Black, White, Asian and Latino & others (4 subgroups).

The total training epoch is 100. In each training epoch, we sample all the subgroups with the same sample size with replacement ($N = 50$). The early stopping strategy is also adopted. The training ($N = 33188$) validation ($N = 3438$) and test set ($N = 9744$) are following the protocol in [40].

We also consider the following sensitive attributes:

1. Religion. The religion includes Christian, Jewish, Muslin and others (such as Hindu, Buddhist, atheist).

2. Gender. The gender includes male, female and others (such as homosexual_gay_or_lesbian, bisexual, transgender, other_gender).

Because toxic comments are marked by multiple annotators, we determine that the comment is toxic if at least half of the annotators mark it. In the implementation, we also adopt the DistilBERT [57] to extract the embedding with dimension $\mathbb{R}^{768}$.

We adopt 4-layers fully connected neural network as the model, where the weights of the model follows the Gaussian distribution $Q_a$ or $Q$. The trade-off coefficient $\lambda$ ranges from $[0.01, 50]$ and we fix $\lambda = 0.6$ in the evaluation. We set all the Monte-Carlo samples as five $N = 5$. More implementation details of experiments and parameter settings can be found in the code.

# H Additional Results in Amazon review

Additional results of each subgroup's fair performance and the probability calibration on the Amazon Review dataset are shown in Fig. 6 and Fig. 7.

# I Additional Results in Toxic Comments

## I.1 Race as sensitive attribute: probability calibration

The additional results of the probability calibration on the Toxic Comments (Race) dataset is shown in Fig. 8.

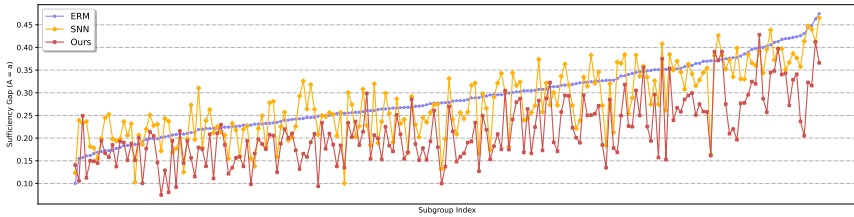

Figure 6: Group sufficiency gap on the specific subgroup $A = a$, which is the gap between $\mathbb{E}[Y|f(X)]$ and $\mathbb{E}[Y|f(X), A = a]$. We visualize the results for the entire users of Amazon Review dataset.

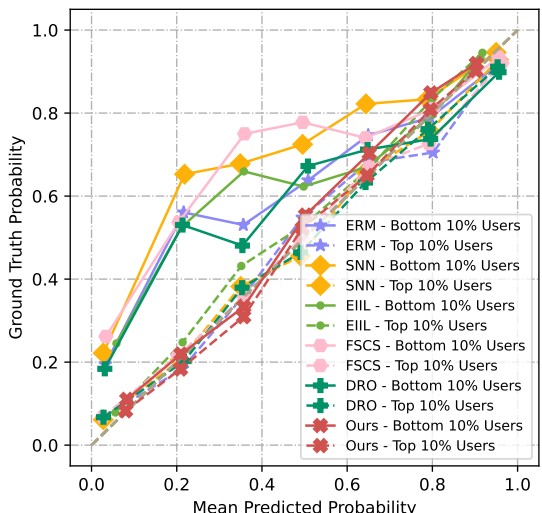

Figure 7: Probability calibration over aggregated users of Amazon Review dataset, where we sort the group sufficiency gap for each subgroup $A = a$, then we visualize the probability calibration curve for the top 10% and bottom 10% users. Since each user has quite limited labels, we aggregate the top 10% and bottom 10% and visualize the probability calibration. The results depict consistently better probability calibration of the proposed approach.

## I.2 Religion as sensitive attribute

The additional results of each subgroup's fair performance and the probability calibration on the Toxic Comments (Religion) dataset are shown in Fig. 9 and Fig. 10.

## I.3 Gender as sensitive attribute

The additional results of each subgroup's fair performance and the probability calibration on the Toxic Comments (Gender) dataset are shown in Fig. 11 and Fig. 12.

It is worth noting that although the group sufficiency in three approaches is quite similar. However, the proposed approach shows a significant better probability calibration than baselines.

## I.4 Results on different subgroup numbers

We visualize the results on different subgroup numbers, shown in Fig. 13. The results still suggest the consistently better results than baselines.

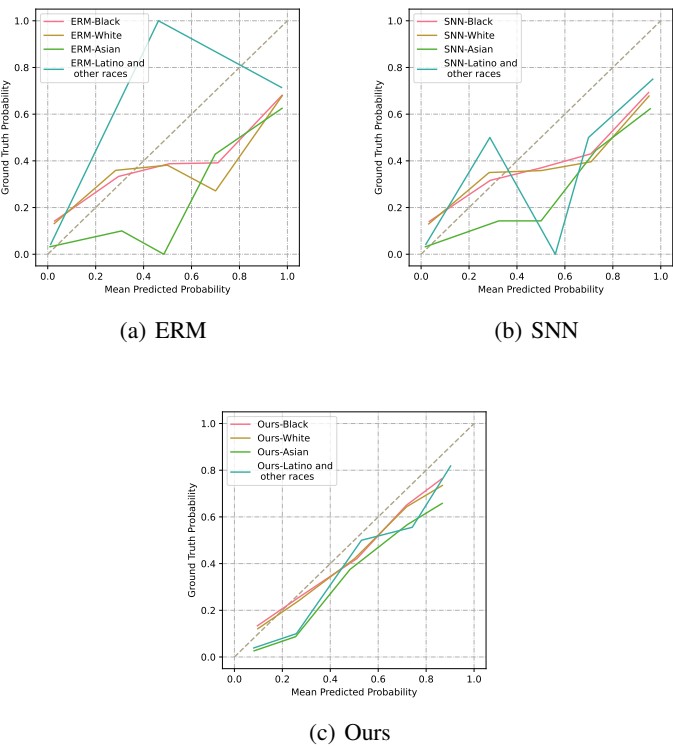

(a) ERM           (b) SNN

(c) Ours

Figure 8: Ablation. Results in Toxic dataset (Race). Visualization of probability calibration for each subgroup $A = a$, i.e. $(f(X), \mathbb{E}[Y|f(X), A = a])$. The proposed approach shows significant improved probability calibration for each subgroup.

## I.5 Additional Results on Adult dataset

We further evaluated the Adult dataset [62], which predicts salary being larger than 50K or not. We treat *gender* as the sensitive attribute and subsample 500 samples for each subgroup. We adopt $\tilde{f}_{\mathbf{w}}$ and $\tilde{f}_{\mathbf{w_a}}$ as the two-layer fully connected neural network, where $\mathbf{w} \sim Q$ and $\mathbf{w}_a \sim Q_a$. All results are repeated 4 times and illustrated in Tab. 1.

Table 1: Results (average and std) in Adult dataset (on %)

| Method | Accuracy | Group sufficiency gap (smaller is better) |
|--------|----------|-------------------------------------------|
| ERM | 83.38 (0.421) | 1.096 (0.182) |
| SNN | 82.62 (0.478) | 1.071 (0.220) |
| EIIL | 83.27 (0.547) | 1.018 (0.239) |
| FSCS | 83.63 (0.704) | 1.271 (0.316) |
| DRO | 83.48 (0.377) | 1.056 (0.151) |
| Ours | 83.33 (0.308) | 0.684 (0.008) |

The result suggests a consistently better group sufficiency with comparable accuracy.

## I.6 Additional Results on vision dataset

We further consider CelebA dataset as a computer vision task [63]. We follow the protocol of [64], which predicts the wavy hair $Y$ in the image $X$. We regard gender as sensitive attribute $A$. We further

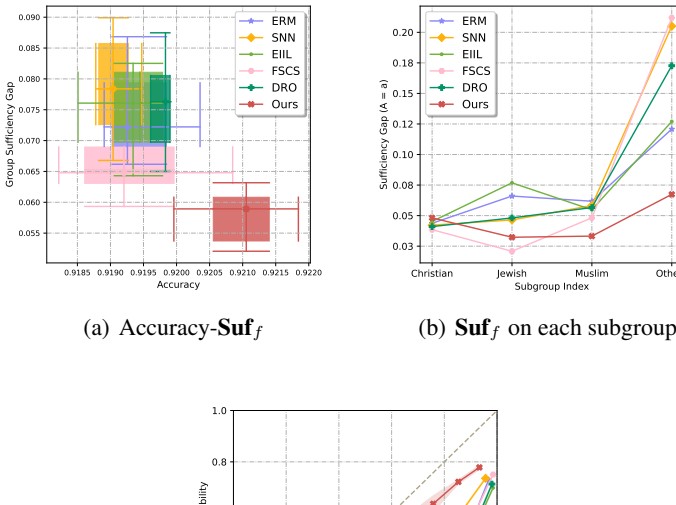

(a) Accuracy-**Suf**$_f$           (b) **Suf**$_f$ on each subgroup

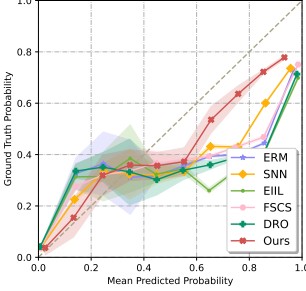

(c) Probability calibration

Figure 9: Results in Toxic dataset (Religion). Boxplot of accuracy and group sufficiency gap **Suf**$_f$ with 5 repeats: median, 75th percentile and minimum-maximum value. (b) Group sufficiency gap on the specific subgroup $A = a$, which is the gap between $\mathbb{E}[Y|f(X)]$ and $\mathbb{E}[Y|f(X), A = a]$. (c) Visualization of probability calibration over 5 repeats with mean and standard deviation. i.e $(f(X), \mathbb{E}[Y|f(X)])$.

adopted Res18 as the backbone and three layers fully-connected (randomized) layers. We sub-sample 200 instances per subgroup and fine tune for maximum 20 epochs. All results are repeated 4 times and illustrated in Tab. 2.

Table 2: Results (average and std) in CelebA dataset (on %)

| Method | Accuracy | Group sufficiency gap (smaller is better) |
|--------|----------|-------------------------------------------|
| ERM | 79.67 (0.40) | 7.13 (0.96) |
| SNN | 79.79 (0.33) | 7.12 (0.89) |
| EIIL | 79.90 (0.30) | 6.23 (1.52) |
| FSCS | 79.33 (0.62) | 7.01 (1.88) |
| DRO | 79.82 (0.27) | 6.68 (1.14) |
| Ours | 79.87 (0.17) | 5.29 (0.67) |

The result also suggests a consistently better group sufficiency with comparable accuracy.

## I.7 Evolution of Q during the training

We visualize the test accuracy and group sufficiency gap of fair predictor $Q$ during the training, shown in Tab. 3. The results are evaluated on the Toxic data with race as the sensitive attribute.

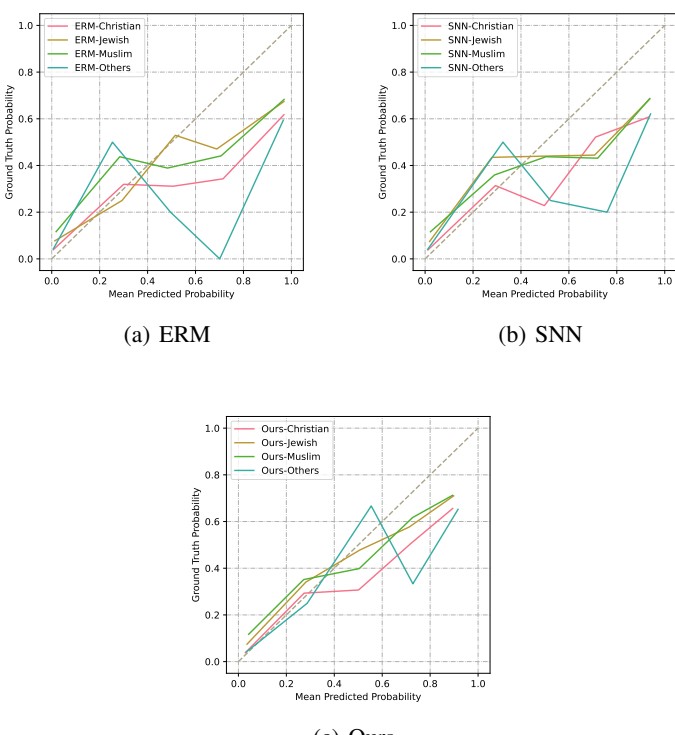

Figure 10: Ablation. Results in Toxic dataset (Religion). Visualization of probability calibration for each subgroup $A = a$, i.e. $(f(X), \mathbb{E}[Y | f(X), A = a])$. The proposed approach shows significant improved probability calibration for each subgroup.

Table 3: Evolution of accuracy and group sufficiency gap during the training (on %)

| Epoch | 0 | 2 | 4 | 6 | 8 | 10 | 12 | 14 |
|---|---|---|---|---|---|---|---|---|
| Suf gap | 12.49 | 14.65 | 11.60 | 7.32 | 6.19 | 5.41 | 5.45 | 5.42 |
| Accuracy | 50.94 | 66.30 | 87.87 | 91.71 | 92.10 | 92.10 | 92.16 | 92.13 |

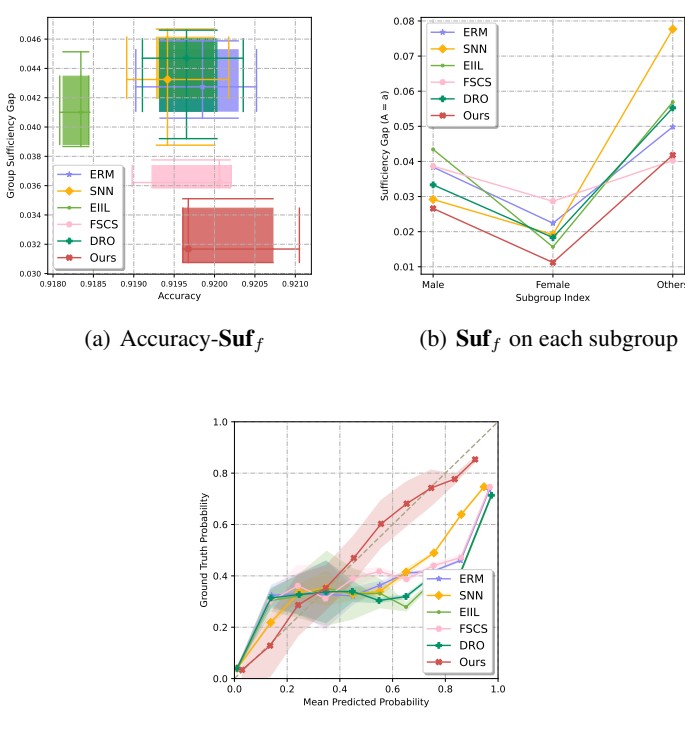

(a) Accuracy-$\mathbf{Suf}_f$

(b) $\mathbf{Suf}_f$ on each subgroup

(c) Probability calibration

Figure 11: Results in Toxic dataset (Gender). Boxplot of accuracy and group sufficiency gap $\mathbf{Suf}_f$ with 5 repeats: median, 75th percentile and minimum-maximum value. (b) Group sufficiency gap on the specific subgroup $A = a$, which is the gap between $\mathbb{E}[Y|f(X)]$ and $\mathbb{E}[Y|f(X), A = a]$. (c) Visualization of probability calibration over 5 repeats with mean and standard deviation. i.e $(f(X), \mathbb{E}[Y|f(X)])$.

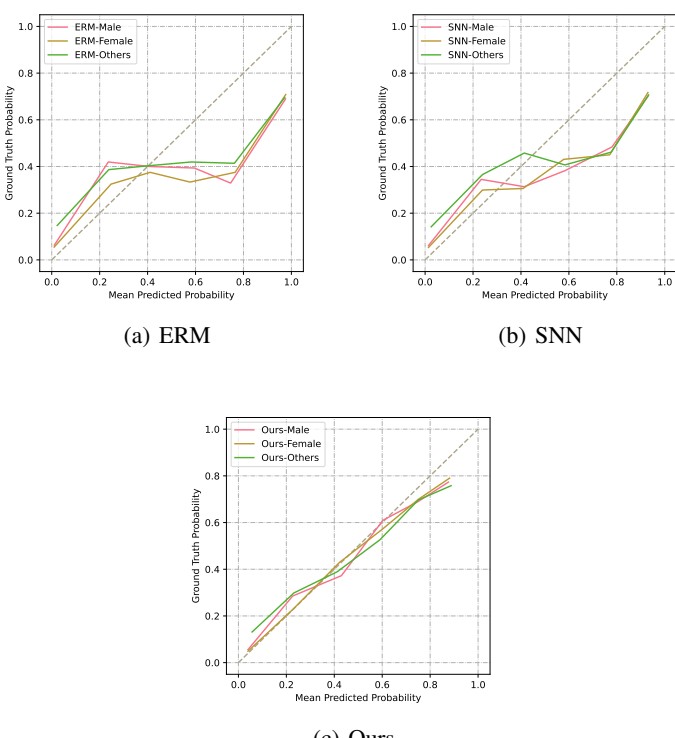

(c) Ours

Figure 12: Ablation. Results in Toxic dataset (Gender). Visualization of probability calibration for each subgroup $A = a$, i.e. $(f(X), \mathbb{E}[Y|f(X), A = a])$. The proposed approach shows significant improved probability calibration for each subgroup.

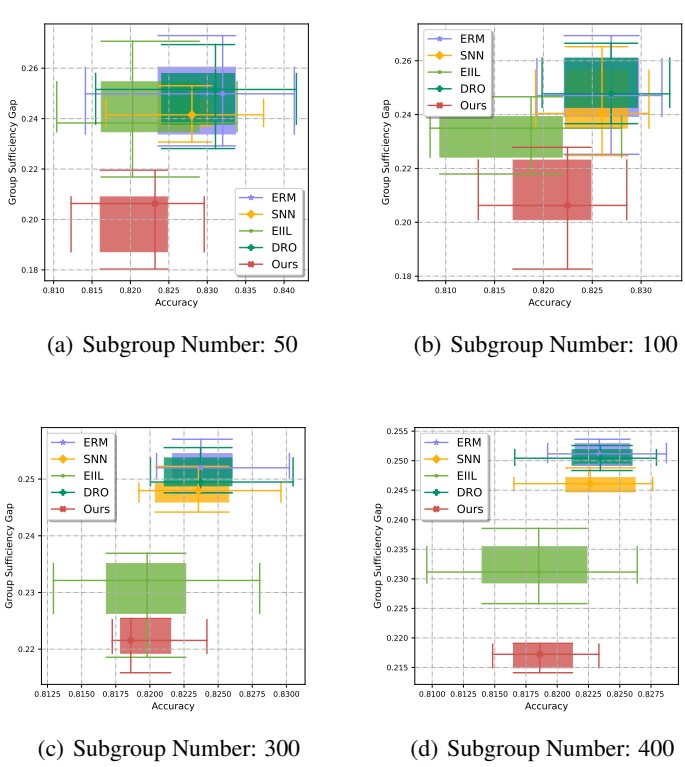

(a) Subgroup Number: 50

(b) Subgroup Number: 100

(c) Subgroup Number: 300

(d) Subgroup Number: 400

Figure 13: Results of different subgroups numbers on Amazon Review Dataset