# OpenReview forum: "On Learning Fairness and Accuracy on Multiple Subgroups"
_NeurIPS.cc/2022/Conference — NeurIPS 2022 Accept_

### Official Review · Reviewer_WyD4 · 2022-07-08

**Rating:** 5
**Confidence:** 3
**Soundness:** 2 fair
**Presentation:** 3 good
**Contribution:** 2 fair

**Summary:**

This paper considers the multiple subgroups scenario in the binary classification task. In order to avoid the inherent trade-off between accuracy and fairness (with fairness criteria such as DP and EO), authors focus on the group sufficiency, which requires that conditional expectation of $Y$ is invariant across different sub-groups. The proposed method is theoretical justified. Several theoretical results including the generalization error and the upper bound on the group sufficiency gap are derived. Based on the upper bound on the group sufficiency gap of the randomized algorithm, a machine learning algorithm is proposed to mitigate the bias of a model while not losing the informativeness. The idea of the novel bilevel optimization algorithm is to train an individual fair predictor by training the subgroup-specific predictors and then aggregating them. The idea of the novel bilevel optimization algorithm is to train an individual fair predictor by training the subgroup-specific predictors and then aggregating them. This allows to achieve superior performance over the other baselines on Amazon review, Toxic comment datasets.

**Questions:**

* Through the proof of proposition 3.1, it was confirmed that the suggested group sufficiency gap can be achieved to zero with $f^{Bayes}$. However, since $\mathbb{E}[Y|f^{Bayes}(X,A_1)] = \mathbb{E}[Y|f^{Bayes}(X,A_2)]$ is not generally satisfied, the invariance of conditional expectation for different subgroups is not established ($A_1$ and $A_2$ denotes two different groups). In other words, even if ${Suf}_{f}$=0, that conditional expectation of ground-truth label given predictor output ($\mathbb{E}[Y|f^{Bayes}(X,A_i), A_i]$) is invariant to the different groups is not guaranteed, which seems undesirable when considering the introduction background of Group sufficiency (line 30-32).
* The assumption that the Bayes-predictors for each group are the same seems to be strong, and the use of different classifiers for each group seems to have induced this phenomenon, what do you think?
* In the algorithm, $Q$ is used as an informative prior, but it seems that $Q$ may not be informative at the beginning of learning. I wonder if there is an experimental result or discussion for that. (Adaptively allocating lambda values according to the epoch may be a way.)
* DRO is a method that considers the worst-group accuracy, so it does not seem surprising that it is not effective in reducing the group sufficiency gap compared to the proposed method. It would be good to add a comment on the background of using DRO as a baseline.

**Limitations:**

It is mentioned that the main theoretical analysis is restricted to the randomized algorithm. It is noted that while Group sufficiency is improved, other fairness criteria such as DP/EO can be deteriorated. Also, authors mentioned that if the variance of Bayes predictors is large, the fair prediction model derived from the proposed algorithm may not that effective.

**Strengths And Weaknesses:**

[Strength]
* The paper is cleary written and the proposed bilevel optimization algorithm based on the theoretical results is original.
* It is quite meaningful that a model can achieve accuracy and fairness simultaneously under the Group sufficiency criterion unlike DP/EO (However, this appears to be achievable only when $f$ is similar)
* It is a great advantage that it can be applied to a variety of datasets, including NLP datasets (in typical fairness methods, there is often no scalability for diverse data types).
* From the group sufficiency of the $A$-group Bayes predictor, it is shown that the model can achieve both informativeness and fairness, simultaneously (Proposition 3.1).
* In theorem 3.2, an upper bound of the group sufficiency gap of any predictor $f$ is derived.
* Based on the theorem 3.2, an upper bound for the randomized algorithm is proposed. (Corollary 4.1).
* In theorem 4.1, the generalization error bound corresponding to the lower-level of the proposed algorithm is drawn.

[Weakness]
* In order for $Q$ to be informative and fair prediction, $f^{Bayes}$ should be similar, which seems to be a strong assumption in practical scenario.
* In algorithm 1, subset $A'$ is sampled for memory saving at each epoch, however, as this type of sampling is quite unfamiliar, it would be better to discuss further.
* Given that it can be applied to various datasets, the additional experimental results on vision datasets will have the effect of highlighting strengths of this paper.
* Experimental results on commonly used datasets in fairness literature such as COMPAS and Adult can be considered. (In these datasets, it seems possible to compare the proposed algorithm with baselines such as FSCS and EIIL.)
* It would be good to conduct an ablation study on using $Q$ as an informative prior.

---

> ### Author Response · Authors · 2022-08-02
> **Response 1/2**
>
> Thanks for your detailed, thoughtful and constructive feedback! We hope the following responses could clarify your confusions, and we are quite happy to provide additional explanations.
>
> > 1. Through the proof of proposition 3.1, it was confirmed that the suggested group sufficiency gap can be achieved to zero. However, E[Y|f^{B}(X,A_1)] = E[Y|f^{B}(X,A_2)] since is not generally satisfied, the invariance of conditional f^B expectation for different subgroups is not established. ( A_1, A_2 and denotes two different groups). In other words, even if suff=0, conditional expectation is not guaranteed invariance to the different groups, which seems undesirable when considering the introduction background of Group sufficiency.
>
> Thanks for your remarks. We think there exist miscommunications about notations and fairness definitions. Specifically:
>
> - Explaining different notations
>      - $f^{B}(X,A=A_1)$ is the Bayes predictor which is obtained from the data in subgroup $A=A_1$.
>      - $f^{B}(X,A=A_2)$ is the Bayes predictor which is obtained from the data in subgroup $A=A_2$.
>      - If we denote $f_1(X) = f^{B}(X,A=A_1)$ and $f_2(X) = f^{B}(X,A=A_2)$, then $E[Y|f^{B}(X, A=A_1)]$ and $E[Y|f^{B}(X, A=A_2)]$ are expressed as:  $E[Y|f_1(X)]$ and $E[Y|f_2(X)]$
>      - $E[Y|f_1(X)]$ means that we use predictor $f_1(X)$ to evaluate the conditional expectation on the **entire data**.
>      - $E[Y|f_1(X)]$ is different from $E[Y|f_1(X),A=A_1]$, which means we we use predictor $f_1(X)$ to evaluate the conditional expectation on the **data in subgroup $A_1$**.
>
> - $E[Y|f^{B}(X,A=A_1)] - E[Y|f^{B}(X,A=A_2)]$ is not a metric of fairness.
>
> 	 Based on the notations,  we have:
>
> 	 $$E[Y|f^{B}(X,A=A_1)] - E[Y|f^{B}(X,A=A_2)] =  E[Y|f_1(X)] -  E[Y|f_2(X)]$$
>
> 	 The above gap measures the difference of conditional expectations. In particular, we adopted two predictors (obtained from $A_1$ and $A_2$) and evaluate conditional expectation on the **entire data** . This gap is not a metric of fairness but reflecting the inherent property of data distribution. Because if $E[Y|f^{B}(X,A_1)] = E[Y|f^{B}(X,A_2)]$, we have $f^{B}(X,A_1) = f^{B}(X,A_2)$, which means there is no distribution shift in these subgroups.
>  - In our paper, the group sufficiency for a predictor $f$ is defined as: $E[Y|f(X)] - E[Y|f(X),A]$: we use one predictor to compute the conditional expectation on the **entire data** and **data in subgroup A**, then computing its gap.
>
>
> > 2. The assumption that the Bayes-predictors for each group are the same seems to be strong, and the use of different classifiers for each group seems to have induced this phenomenon, what do you think?
>
> Thanks for your remark. We would like to explain the similar A-Bayes predictor for each subgroup is not a strong assumption in our context, and this assumption is also not caused by using different subgroup classifiers. Specifically,
>
> - **Similar A-Bayes predictor is the inherent assumption about data distribution and independent of the algorithm.** In fact, the A-Bayes predictor  $E[Y|X,A]$, is the conditional expectation w.r.t. the data-generation distribution.
>
> - **Learning informative predictor under completely different A-Bayes predictors is impossible.**
> We aim to understand when we could simultaneously achieve informative (with good accuracy) and fair predictions, where each subgroup contains *limited* samples. If A-Bayes predictors $E[Y|X,A]$ are completely different for different subgroup $A$, all baselines and our approach could not learn an informative predictor.
> For example, consider two subgroups $A_0, A_1$, for a given input $X_0$, $Y=0$ in subgroup $A_0$  and $Y=1$ in subgroup $A_1$, i.e., $E[Y|X=X_0,A=A_0] = 0$, $E[Y|X=X_0,A=A_1] = 1$.
>  In this case, we could not learn a classifier $f(X)$ to simultaneously correctly predict $f(X_0)$ from subgroup $A_0$ and $A_1$ since they are completely different.
>
> - **Using different classifiers in the lower-level optimization is motivated by the Theorem.**
> Based on Theorem 3.2, to control group sufficiency gap, we need to find a classifier $f$ to be close to all the A-Bayes optimal predictors. In practice we further train subgroup specific classifier to approximate A-Bayes optimal predictors, which is learned by limited subgroup data and fair-prior classifier.

---

> > ### Author Response · Authors · 2022-08-02
> > **Response 2/2**
> >
> > > 3. In algorithm 1, subset is sampled for memory saving at each epoch, however, as this type of sampling is quite unfamiliar, it would be better to discuss further.
> >
> > Thank you for your thoughtful remarks. According to Theorem 4.1, all subgroup specific predictors is learned to update the fair predictor when the subgroup number is modest. In contrast, if we face many subgroups  (e.g., 200), due to memory constraints, it would be difficult to save entire models in the lower-level if the subgroup number is large.  As a result, we used a simple sampling strategy to reduce memory complexity: we only randomly sample a subset of groups in each training epoch. In this epoch, we then trained the corresponding subgroup-specific models as well as the fair model.
> >
> > However, if only few subgroups (e.g., 2) are chosen, learning an informative prior $Q$ will be difficult and unstable. In practice, we usually select a relatively large subset of groups to ensure the training stability.
> >
> > > 4. In the algorithm, Q is used as an informative prior, but it seems that may not be informative at the beginning of learning. I wonder if there is an experimental result or discussion for that.
> >
> > Thanks for your suggestion. We use a common *alternating optimization* to solve the bi-level objective, and $Q$ gradually improves. We think that the convergence behaviour can be demonstrated by considering proper optimization assumptions. We also show the test accuracy and group sufficiency gap at different training epochs. The findings suggested that accuracy and fairness were gradually improving.
> >
> >
> >  Epoch    | 0        | 2        | 4        | 6        | 8        | 10       | 12       | 14       |
> > |----------|----------|----------|----------|----------|----------|----------|----------|----------|
> > | Suf Gap (\%) | 12.49 | 14.65| 11.60 | 7.31 | 6.19 | 5.40 | 5.44 | 5.42  |
> > | Accuracy (\%)| 50.94 | 66.30 | 87.87 | 91.71 | 92.09 | 92.09 | 92.16 | 92.13 |
> >
> > > 5. DRO is a method that considers the worst-group accuracy, so it does not seem surprising that it is not effective in reducing the group suffciency gap compared to the proposed method. It would be good to add a comment on the background of using DRO as a baseline.
> >
> > We agree with your viewpoint in DRO, and we will revise the manuscript accordingly.
> >
> > >6. Given that it can be applied to various datasets, the additional experimental results on vision datasets will have the effect of highlighting strengths of this paper.
> >
> > Thanks for your suggestions. We added a vision task (CelebA data) and experimental details are in Appendix. The  results are as follows:
> >
> >  Method |       Suf Gap (\%)      |       Accuracy (\%)     |
> > |:------:|:-------------------:|:-------------------:|
> > |   ERM  |   7.13 ± 0.96   |   79.67 ± 0.40   |
> > |   SNN  |   7.12 ± 0.89   |   79.79 ± 0.33   |
> > |  EIIL  |   6.23 ± 1.52   |   79.90 ± 0.30  |
> > |  FSCS  |   7.01 ± 1.88   |   79.33 ± 0.62   |
> > |   DRO  |   6.68 ± 1.14   |   79.82 ± 0.27   |
> > |  Ours  |   5.29 ± 0.67   |   79.87 ± 0.17   |
> >
> > > 7. Experimental results on commonly used datasets in fairness literature such as COMPAS and Adult can be considered.
> >
> > Thank you for your suggestion. The Adult dataset has been added (the experimental details are in the Appendix),
> >
> > | Method |        Accuracy (\%)     |          Suf Gap (\%)      |
> > |:------:|:---------------------:|:------------------------:|
> > |   ERM  |   83.38 ± 0.4210   |    1.096 ± 0.1819    |
> > |   SNN  |   82.62 ± 0.4783   |    1.071 ± 0.2203    |
> > |  EIIL  |   83.27 ± 0.5473   |    1.018 ± 0.2387    |
> > |  FSCS  |   83.63 ± 0.7041   |    1.271 ± 0.3162    |
> > |   DRO  |   83.48 ± 0.3767   |    1.056 ± 0.1512    |
> > |  Ours  |   83.33 ± 0.3078   |    0.684 ± 0.0809    |

---

> > > ### Comment · Reviewer_WyD4 · 2022-08-06
> > > **Something confused**
> > >
> > > Thanks for the rebuttal.
> > > In Question 1, what i meant was as follows:
> > >
> > > * As an example, consider $A_i \in \mathcal{A}$, where $|\mathcal{A}|=4$.
> > >
> > > * From line 30-32, I firstly noted that the fairness criteria this paper targets is the group sufficiency (equalization of conditional expectation of ground-truth label given predictor output, i.e., $\mathbb{E}[Y|f(X), A_i] = \mathbb{E}[Y|f(X), A_j]$ for all $A_i, A_j \in \mathcal{A}$).
> > >
> > > * In the proof of Proposition 1, authors showed that $f^{Bayes}$ satisfies $\mathbb{E}[Y|f^{Bayes}(X, A_i), A_i] = \mathbb{E}[Y|f^{Bayes}(X, A_i)]$ for all $A_i \in \mathcal{A}$, separately.
> > >
> > > * With these four $f^{Bayes}$ classifiers, the group sufficiency gap (defined in Definition 2.2) becomes zero even if the $f^{Bayes}$ classifiers are different.
> > >
> > > * However, this does not mean ‘equalization of conditional expectation of ground-truth label given predictor output’ (i.e., $\mathbb{E}[Y|f^{Bayes}(X, A_i), A_i] = \mathbb{E}[Y|f^{Bayes}(X, A_j), A_j]$ for all $i,j \in \mathcal{A}$) in general.
> > >
> > > * Thus, I wonder if the A-group Bayes predictor indeed achieves the fairness criterion mentioned in line 30-32.
> > >
> > > * (If we use only one predictor $f$ for all the subgroups, ‘$Suf_f = 0$’ implies $\mathbb{E}[Y|f(X), A_i] = \mathbb{E}[Y|f(X), A_j]$ for all $i,j \in \mathcal{A}$, but this is not the case.)
> > >
> > > If I miss something, please let me know.

---

> > > > ### Author Response · Authors · 2022-08-07
> > > > **Thanks for your comments!**
> > > >
> > > > We appreciate your thoughtful question, this isn't just a boilerplate.  Indeed it helped to improve the rigour of our paper. We hope our responses are helpful, and we are happy to provide further explanations.
> > > >
> > > > > As an example, consider  Ai∈A, where  |A|=4.
> > > >
> > > > > From line 30-32, I firstly noted that the fairness criteria this paper targets is the group sufficiency (equalization of conditional expectation of ground-truth label given predictor output, i.e.,  $E[Y|f(X),A_i]=E[Y|f(X),A_j]$  for all  $A_i,A_j∈A$).
> > > >
> > > >  > In the proof of Proposition 1, authors showed that  fBayes  satisfies  $E[Y|f^B(X,A_i),A_i]=E[Y|f^{B}(X,A_i)]$  for all  $A_i∈A$, separately.
> > > >
> > > > We agree with your thoughts. In our definition, $f(X)$ is a single function, where the input is only $X$ without explicit information about group $A$.
> > > >
> > > > >  1. With these four  $f^{B}$  classifiers, the group sufficiency gap (defined in Definition 2.2) becomes zero even if the  $f^{B}$  classifiers are different.
> > > >
> > > > We would like to explain that the definition of group sufficiency gap is also defined as $E_{A,X}|E[Y|f(X)] - E[Y|f(X),A]|$, where $f(X)$ is a *single* predictor with an input $X$. That means, we could not compute the group sufficiency gap by using 4 different A-Bayes group predictors.  If we adopt any one of these 4 predictors to compute the group sufficiency gap, it does not equal to zero.
> > > >
> > > > >  2.  However, this does not mean ‘equalization of conditional expectation of ground-truth label given predictor output’ (i.e.,  $E[Y|f^{B}(X,A_i),A_i]=E[Y|f^{B}(X,A_j),A_j]$  for all  $i,j∈A$) in general.
> > > >
> > > > We also would like to explain that $E[Y|f^{B}(X,A_i),A_i]=E[Y|f^{B}(X,A_j),A_j]$  does not suggest the definition of  group sufficiency, because $f^{B}(X, A=A_i) \neq f^{B}(X, A=A_j)$ are essentially two different predictors, whereas group sufficiency is defined on the single predictor $f$ that does not change across group.
> > > >
> > > > >  3.  Thus, I wonder if the A-group Bayes predictor indeed achieves the fairness criterion mentioned in line 30-32.
> > > >
> > > > We are sorry for the confusion in describing Proposition 3.1. Indeed it proved that **given each subgroup $A=A_i$**, we have $E[Y|f^{B}(X,A_i), A=A_i] = E[Y|f^{B}(X,A_i)]$ of the same A-Bayes predictor $f^{B}(X,A_i)$. But this does not imply we have a *global and single* predictor $f(X)$ to satisfy the group sufficiency.  We will illustrate this concept explicitly in the manuscript.
> > > >
> > > > We would like to express the intuition in Proposition 3.1, which shows that **given a subgroup** $A_i$, $f^{B}(X,A_i)$ could achieve the equal conditional expectation $E[Y|f^{B}(X,A_i), A=A_i] = E[Y|f^{B}(X,A_i)]$ and optimal predictor within subgroup $A_i$. Further $f^{B}(X,A=A_i)$ is not optimal in other subgroups $A_j$, $i\neq j$.
> > > >
> > > > In our algorithm and Theorem 3.2, we aim to only use a *single* predictor $f$ to minimize group sufficiency gap, which is aligned with all definitions.  Inspired by proposition 3.1, $f$ should be **as close to all the A-group Bayes predictors (also optimal) as possible** such that $E[Y|f(X),A]\approx E[Y|f(X)]$. Theorem 3.2 further formally justified this fact: the group sufficiency upper bound of $f$ is controlled by the closeness of A-group Bayes predictors.
> > > >
> > > > Thanks again for your insightful comments!

---

> > > > > ### Comment · Reviewer_WyD4 · 2022-08-08
> > > > > **Raised score to 5**
> > > > >
> > > > > Thank you for the detailed comments. Based on your comment, I was able to understand the paper better. There are some remaining questions and suggestions, but, since they are rather minor, I believe the authors will reflect in the final version, and I modify the score.
> > > > >
> > > > > [Suggestions]
> > > > > * line 116 & line 120: “group sufficiency” → “group sufficiency w.r.t $A$” (considering 3.1 section in [21] and Definition 2.1 in this paper)
> > > > > * $f^{Bayes}(X,A)$ → $ f_{A}^{Bayes}(X) $ thorough out the paper:
> > > > > To avoid misunderstanding by readers, I recommend that $f^{Bayes}_{A}(X)$ notation to be considered instead of $f^{Bayes}(X,A)$ (In my opinion, this allows to emphasize that $f^{Bayes}$ varies by group in Theorem 3.2).
> > > > >
> > > > > [Questions]
> > > > > * line 105-107: Similar to Group sufficiency, DP & EO can also encourage two objectives under certain settings. In my opinion, it would be good to modify the nuance slightly weakly. How do you think ?
> > > > > 1. EO allows a perfectly accurate predictor (Theorem 5.6 & introduction paragraph in [8]).
> > > > > 2. DP allow a perfectly accurate predictor when base rates are equal across groups.
> > > > >
> > > > > * It seems that If there is no ‘uniform distribution’ assumption in Theorem 3.2, the form of Theorem 3.2 and thus Corollary 4.1 will be only slightly different. As this extension is thought to be more general, I want to know the roles of 'uniform distribution' throughout the paper (I guess that algorithmic design is also considered for the introduction of ‘uniform distribution’. Or, Is it introduced for just proof ?). Please let me know.

---

> > > > > > ### Author Response · Authors · 2022-08-08
> > > > > > **Thanks for your suggestions**
> > > > > >
> > > > > > We deeply appreciate your engagements and suggestions. Our thoughts are as follows:
> > > > > >
> > > > > > > -   line 105-107: Similar to Group sufficiency, DP & EO can also encourage two objectives under certain settings. In my opinion, it would be good to modify the nuance slightly weakly. How do you think ?
> > > > > >
> > > > > > > 1.  EO allows a perfectly accurate predictor (Theorem 5.6 & introduction paragraph in [8]).
> > > > > > > 2.  DP allow a perfectly accurate predictor when base rates are equal across groups.
> > > > > >
> > > > > > We agree with your thoughts about EO and DP, and we will fix them in the next version.
> > > > > >
> > > > > > > -   It seems that If there is no ‘uniform distribution’ assumption in Theorem 3.2, the form of Theorem 3.2 and thus Corollary 4.1 will be only slightly different. As this extension is thought to be more general, I want to know the roles of 'uniform distribution' throughout the paper (I guess that algorithmic design is also considered for the introduction of ‘uniform distribution’. Or, Is it introduced for just proof ?). Please let me know.
> > > > > >
> > > > > > Our corollary and algorithm both considered uniform distribution, and we could extend them to the non-uniform distribution. In this case, the upper-level loss will be expressed as the weighted sum by $P(A=a)$ (the original is uniform).
> > > > > >
> > > > > > Specifically, the distribution on A reflects the *ground-truth* probability of subgroups, and it is generally regarded as an *assumption* in different contexts. For example, we assume P(A='male')=P(A='female')=0.5 in the ground-truth distribution. And this is not the observational distribution, the observational dataset could be biased.
> > > > > >
> > > > > > If we assume a non-uniform distribution, such as P(A='male') = 0.8, P(A='female') = 0.2, the weights in female will be significantly reduced. We think this is not a proper assumption in terms of fairness, so we assume all subgroup probabilities are equal.

---

### Official Review · Reviewer_Qbu1 · 2022-07-11

**Rating:** 7
**Confidence:** 3
**Soundness:** 3 good
**Presentation:** 3 good
**Contribution:** 3 good

**Summary:**

This paper presents a method for learning a fair predictor which is designed for scenarios where the data contains multiple subgroups, each with limited number of samples. This method formulates the subgroups fair predictor question as a bilevel objective.

post-rebuttal
Thank you for the clarifications. I will raise my score to 7.

**Questions:**

(1) Need optimal $\lambda$ to be tuned by hand for each dataset?
(2) As I described in the weakness part, I think it's not a fair comparison in Figure 3(a). Could you please show the group sufficiency gap for each dataset with the same accuracy?
(3) How does the number of subgroups influence the final result.

**Limitations:**

The author didn't described the limitations of this work.

**Strengths And Weaknesses:**

 Strengths:
(1) The paper proposed a novel method and provides solid prof.
(2) The paper's framework seems technically sound. The gradient estimates seem reasonable to me.
(3) The paper focuses on group sufficiency gap to measure fairness
Weaknesses:
(1) For Figure 3(a), though the method could achieve a lower group sufficiency gap, the accuracy is worse. I think this result is not enough to show the benefit because of the trade off between accuracy and fairness.
(2) The bilevel optimization idea is not very novel as there are some existing works such as Ditto[1].
[1] Tian Li, Shengyuan Hu, Ahmad Beirami, and Virginia Smith. Ditto: Fair and robust federated learning through personalization. In International Conference on Machine Learning, pages 6357–6368. PMLR, 2021.

---

> ### Author Response · Authors · 2022-08-02
> **Response**
>
> Thanks for your detailed, thoughtful and constructive feedback! We hope the following responses could clarify your confusions, and we are quite happy to provide additional explanations.
>
> > 1.For Figure 3(a), though the method could achieve a lower group sufficiency gap, the accuracy is worse. I think this result is not enough to show the benefit because of the trade off between accuracy and fairness. As I described in the weakness part, I think it's not a fair comparison in Figure 3(a). Could you please show the group sufficiency gap for each dataset with the same accuracy
>
> Thanks for your remark. Because the fairness and accuracy are evaluated on the *test-set* in Fig 3(a) (Amazon review), we could not control the same test accuracy for all baselines. Specifically, the proposed approach's accuracy ranges from [0.816,0.823], whereas other baselines generally range from [0.817,0.827], indicating that the accuracy is generally comparable with other baselines. The group sufficiency gap, on the other hand, shows a clear improvement: [0.203, 0.221] (ours), [0.228,0.250] (EIIL), and [0.248,0.262] (other baselines). Thus our conclusion is that group sufficiency has been consistently improved, with almost no loss of accuracy.
>
>
> > 2. The bilevel optimization idea is not very novel as there are some existing works such as Ditto[1]
>
> Thanks for the remark. We agree that practical bi-level optimization is merely a tool that is used in various forms such as min-max optimization, reweighting, etc. Our *key* point is to show the connection between the group sufficiency and our proposed bi-level optimization objective. Moreover, we further rigorously justify the role of lower- and upper-level losses in our framework, which control group sufficiency and generalisation error, respectively.
>
> We also included a related work on bi-level optimization in appendix A.1. We hope that this will enable a better understanding of bi-level optimization (such as the work mentioned above) and fairness.
>
> > 3. $\lambda$ Need optimal to be tuned by hand for each dataset
>
> Thanks for your comments. In fact the optimal $\lambda$ theoretically depends on the dataset. At the same time, our theoretical results proposed a principled way to fast determining the optimal coefficient. The optimal theoretical coefficient, according to Theorem 4.1 (Page 5, footnote 2), is $L\sqrt{A/m}$.
>
> Where $L$ is the loss w.r.t. the data, $A$ is the number of subgroup and $m$ could be viewed as the subgroup samples. If we consider Amazon review dataset, and assume loss $L=1$, $A=200$ and $m=150$, then we could compute  $\lambda \approx 1.22$. In practice, the analysis in Fig 5 illustrated results on different $\lambda$, showing the optimal $\lambda = 1$, which is close to our theoretical analysis.
>
> > 4. How does the number of subgroups influence the final result.
>
> Thanks for your suggestions. We *added additional experimental results* by changing the subgroup numbers in Amazon review dataset (Appendix A.2, with subgroup number = 50,100,300,400). The results indicate that the proposed framework performs consistently better than baselines.

---

> ### Author Response · Authors · 2022-08-09
> **Thanks for your feedback**
>
> Thanks for your feedback ! We will include your suggestions in the next version.

---

### Official Review · Reviewer_2NvT · 2022-07-11

**Rating:** 8
**Confidence:** 5
**Soundness:** 4 excellent
**Presentation:** 4 excellent
**Contribution:** 3 good

**Summary:**

This paper discussed learning group sufficiency and accuracy for multiple subgroups. More specifically, it can have many subgroups and limited samples for each subgroup, which is a practically realistic and challenging task. The authors proposed a bi-level optimization to learn the fair predictor, and further derived several theoretical results to justify that both group sufficiency and accuracy could be controlled. The empirical performance demonstrated better fairness and accuracy of the proposed approach.

**Questions:**

It would be great if the author could address the questions in weakness and comments.

**Limitations:**

This paper discussed the limitations in the checklist.

**Strengths And Weaknesses:**

Strength:
1.	The fairness topic discussed in this paper is interesting and important: when and how we could simultaneously achieve fairness (groups sufficiency) and accuracy (small generalization error).
2.	Based on the theoretical analysis, this paper derived a novel fair learning approach. Besides, extensive empirical results demonstrated the superiority of the proposed approach.
3.	In general, the paper is clearly written and well presented.
Weaknesses and comments:
1.	This paper mainly focused on group sufficiency as the fairness metric. Is it possible to derive similar results under criteria of demographic parity or equalized odds? What are the potential challenges for other fair metrics? Under these settings, is it still possible to achieve both fairness and accuracy for many subgroups?
2.	The regularization coefficient $\lambda$ seems to have a joint optimal value in 0.1-2. Could you elaborate more on why both fairness and accuracy drop when $\lambda$ is large?
3.	Is it possible to assume the general gaussian distribution rather than isotropic gaussian in the proposed algorithm? What is the difference?
4.	Can the proposed theoretical analysis be extended for a regression or segmentation task? For example, could we obtain the same results as the classification task?
5.	Could you explain a bit more on the intuition of group sufficiency? Is there any relation to the well-known sufficient statistics?
        Other comments:
1.	Could we extend the protected feature $A$ to a vector form? For instance, $A$ represents multiple attributes.
2.	In the Introduction part, the authors introduced a medical therapy instance to present the importance of group sufficiency. Could you explain a bit more about the difference between sufficiency and DP/EO metrics in the real-world application scenarios?
3.	In line 225 and line 227, the mathematical expression of gaussian distribution is ambiguous.
4.	In section 4.4, it mentions the utilization of Monte-Carlo sampling method. I am curious about the influence of different sampling numbers.


================================================
Thanks the effort from the authors, and I am satisfied with the rebuttal. I would like to raise my score to 8.

---

> ### Author Response · Authors · 2022-08-02
> **Response 1/2**
>
> Thanks for your detailed, thoughtful and constructive feedback! We hope the following responses could clarify your confusions, and we are quite happy to provide additional explanations.
>
> > 1. This paper mainly focused on group sufficiency as the fairness metric. Is it possible to derive similar results under criteria of demographic parity or equalized odds? What are the potential challenges for other fair metrics? Under these settings, is it still possible to achieve both fairness and accuracy for many subgroups?
>
> Thank you for your remarks. Demographic parity (DP) and equalized odds (EO) have different behaviors than sufficiency.  In fact, if sensitive attribute $A$ is dependent of $Y$, then DP and accuracy are impossible to hold at the same time.  According to (Liu et al. 2019), there is a lower bound on the accuracy of EO if $A$ is dependent of $Y$. Thus it is difficult to simultaneously achieve accuracy and EO/DP in the general scenarios. When $A$ is independent of $Y$, the EO/DP can be achieved simultaneously while maintaining good accuracy. While it remains unclear how to derive a theoretical principled approach to achieving these two goals simultaneously.
>
> Besides, group sufficiency requires more mild assumptions than EO/DP to achieve fair and accuracy trade-off, where group sufficiency does not require the independence between $A$ and $Y$.
>
> > 2. The regularization coefficient seems to have a joint optimal value in 0.1-2. Could you elaborate more on why both fairness and accuracy drop when is large
>
> Thanks for your remark. When $\lambda$ is large, the regularization term (KL divergence between fair prior predictor and subgroup specific predictor) dominates the lower-level loss. The group specific predictor would then be unable to learn from each subgroup data because it is influenced by fair prior predictor.
>
> As a result, the prior fair predictor could not then capture the shared information of all subgroup specific predictors at the upper-level, resulting in a low accuracy.
> Furthermore, even with a random/non-informative predictor, a small sufficiency gap could not be guaranteed (e.g, the counterexample in Appendix B). As a result, both accuracy and fairness suffer.
>
> > 3. Is it possible to assume the general gaussian distribution rather than isotropic gaussian in the proposed algorithm? What is the difference?
>
> Thanks. We think the isotropic gaussian distribution is used because (1) it reduces computational complexity. In fact, if the parameter space is $d$ dimensional, an isotropic gaussian requires only $d$ dimensions, whereas a full gaussian matrix requires $d^2$ dimensions. (2) A large parameter space on the covariance matrix will result in *overfitting*. We select the isotropic gaussian distribution based on these criteria.
>
> > 4.Can the proposed theoretical analysis be extended for a regression or segmentation task? For example, could we obtain the same results as the classification task?
>
> Actually we think the proposed approach can be extended to regression and segmentation. Because we are approximating the A-Bayes predictor, which is optimal classifier under cross-entropy.  In fact, the proposed algorithm can be applied in various kinds of losses and network structures.
>
> > 5.  Could you explain a bit more on the intuition of group sufficiency? Is there any relation to the well-known sufficient statistics?
>
> Thanks for your remarks. Intuitively, group sufficiency implies that, given the same predicted output, the expectation of ground-truth label will be the same across all subgroups. In the health example, if the health-risk score is identical across races, the conditional expectation of ground-truth health risk should be identical as well.
>
> While we think that sufficiency rule is different from sufficient statistics. In fact, the sufficient statistics is to find statistics that are conditionally independent of a given parameter $\theta$. The sufficiency rule, like the concept in sufficient statistics, encourages conditional independence.
>
>
> > 6.Could we extend the protected feature to a vector form? For instance, represents multiple attributes.
>
> We are currently focusing on a single protected attribute such as gender, but it could be practically extended to multi-protected attribute settings. Consider gender and race, then we can create subgroups solely on gender and race (i.e., group 1 = male, group 2 = female, group 3 = black, group 4 = white, with some overlap between the subgroups). Then we could run our framework by learning four subgroup-specific predictors and then learning a fair-informative predictor at upper-level. Because the fair predictor enforces all identical subgroup optimal predictors, it could achieve fair results across gender and race. Simultaneously, we think this is merely an empirical extension; the exact theoretical assumptions for the provable fairness remain opening.

---

> > ### Author Response · Authors · 2022-08-02
> > **Response 2/2**
> >
> > > 7. In the Introduction part, the authors introduced a medical therapy instance to present the importance of group sufficiency. Could you explain a bit more about the difference between sufficiency and DP/EO metrics in the real-world application scenarios.
> >
> > Thanks for highlighting the implications of group sufficiency. Indeed, in health-related applications, EO/DP would generally enforce the (marginal or conditional) output score, which is undesirable in several context of healthcare. Disease distributions vary across different subgroups, thus enforcing EO/DP essentially degrades the prediction performance.  As a result, sufficiency may be a more appropriate metric to ensure fairness and accuracy.
> >
> >
> > > 8. In line 225 and line 227, the mathematical expression of gaussian distribution is ambiguous.
> >
> > Thanks for pointing this out. In fact, we assume that the distribution of neural-network parameters is isotopic gaussian. The mean and variance of each neuron then follow a one-dimensional gaussian distribution. The product of each neuron will then be the joint distribution of the neural network parameters.
> >
> > > 9. In section 4.4, it mentions the utilization of Monte-Carlo sampling method. I am curious about the influence of different sampling numbers.
> >
> > Thank you for your remarks. We conduct ablation studies, where the results show that MC samplings (>=5) has no significant effect on the results.
> >
> > |        MC samplings                |     1    |    2    |     5    |     8    |
> > |:--------------------------------:|:--------:|:-------:|:--------:|:--------:|
> > |             Accuracy (\%)            |  89.39  |  91.78 |  92.21  |  91.83  |
> > |              Suf Gap (\%)           |  8.391 |  6.09 |  5.742 |  6.975 |
> > | Training Time (per epoch/second) | 0.472512 | 0.55203 | 0.827588 | 1.106873 |

---

> ### Author Response · Authors · 2022-08-08
> **Thanks for your feedback**
>
> Thanks for your feedback ! We will include your suggestions and comments in the next version.

---

### Official Review · Reviewer_wL7w · 2022-07-13

**Rating:** 7
**Confidence:** 4
**Soundness:** 3 good
**Presentation:** 3 good
**Contribution:** 3 good

**Summary:**

This paper presents an approach for learning predictors that have a small 'sufficiency' gap. Here the sufficiency gap is a difference that measures how invariant a predictor is across different subgroups.  The paper gives a theorem that bounds the sufficiency gap of a predictor in terms of the bayes optimal classifier. The proposed bound is then specified to the learned/randomized predictor in a separate corollary leading to 1) an optimization term and 2) an approximation term. Finally, the paper gives a bilevel optimization formulation for learning a predictor that has small sufficiency gap. In the lower level objective, the goal is to learn group specific predictors that while the upper level loss updates a global objective that tries to make the predictor as 'close' as possible to each of the subgroup predictors learned by the lower level problem. The bilevel formulation is compared to other baselines on different datasets where it has favorable performance.

**Questions:**

The main questions that I have a listed in the weakness section above. The ones I would like to hear the authors opinion on are 1) Challenge with different structural groups and 2) Feasibility.

**Limitations:**

An explicit discussion of limitations is not in the draft; perhaps this can be included in the next iteration.

**Strengths And Weaknesses:**

## Strengths
- **Studies Important Problem & Proposes an Interesting Solution**: The paper studies an important but open problem of learning predictors that have low sufficiency gap. The bilevel formulation presented is also interesting and is a relatively new way to approach the problem.

- **Interesting Generalization Error Bound**: Theorem 4.1 gives a generalization error bound of learning group-based predictors, which is also an interesting result.

- **Empirical Performance**: The results and comparison do seem to show that the bilevel formulation here either compares favorably or outperfoms other methods.


## Weaknesses
None of the weaknesses here are really disqualifying, but could be easily addressed in the paper.

- **Limited Samples**: The paper motivation of the paper is that there are only limited samples in each subgroup; however, how can there only be limited samples if one still has to learn a predictor for that subgroup? In practice, I don't think this approach works with only a limited number of samples. However, this issue is a matter of semantics.

- **Feasibility**: It is unclear how such group based approaches will scale with the number of groups. If $\vert \mathcal{A} \vert = 100$, is this approach even feasible? There is an assumption here that the $ \mathcal{A}$ partitions the available samples, but in practice you can have a sample be the intersection of two protected attributes, i.e., race and gender. In practice, poor performance often persists for the groups that happen to be intersection of protected attributes.

- **Link From Theorems to Bilevel Formulation**: It is not clear to me how the propositions and theorems in the paper suggest a bilevel formulation. It is still possible to relax the bilevel problem into a single objective with several penalties for each group. Not sure what in the theorems really necessitates a bilevel formulation.


- **Take away from the theorems**: Several of the theorems are in terms of the bayes optimal classifier for either a group or all data. Since the bayes classifier is usually infeasible to compute in practice, it is a bit hard to translate these insights to practice.


- **Challenge with different structural groups**: I wonder if the authors can speak to issues that occur when, say two groups just have different structural properties. Consider a 1-d case where a group of points have two regression parameters with positive slope, but the other groups has true parameters with negative slope. I am not sure how the formulation will handle that. Perhaps this is a limitation of the current setup?

- **Missing Literature on Bilevel Optimization**: There is an emerging rich literature on bilevel optimization that this paper did not reference or discuss. Once can often solve these problems with implicit differentiation or unrolling(as is done here). A discussion of these points is needed here.

---

> ### Author Response · Authors · 2022-08-02
> **Response 1/2**
>
> Thanks for your detailed, thoughtful and constructive feedback! We hope the following responses could clarify your confusions, and we are quite happy to provide additional explanations.
>
> >1. Limited Samples: The paper motivation of the paper is that there are only limited samples in each subgroup; however, how can there only be limited samples if one still has to learn a predictor for that subgroup? In practice, I don't think this approach works with only a limited number of samples.
>
> - Thanks for your comments. We agree that **without any assumptions**, it is impossible to learn a fair and informative predictor from limited samples for each subgroup.  As a result, additional assumptions will be considered, such as the data generation distribution of each subgroup $D(Y|X,A)$ being similar. Then we could learn shared and fair (in terms of group sufficiency) model from a large number of subgroups.
>
> - **Practical implications**   We empirically evaluated sentiment classification (NLP task) of multiple clients (protected feature $A$) in Amazon review dataset, each client with only limited reviews. If we simply combine all client's data, the ERM algorithm exhibits biased prediction for some clients. In contrast, our framework ensures both fair and informative for all clients. In fact, we assume the the relationship between sentiment $Y$ and review text $X$: $D(Y|X,A)$ will be similar across clients ($A$). For instance, if X = "good product," Y will be positive by 5 star or  4 star depending on the client.
>
> > 2.Feasibility: It is unclear how such group based approaches will scale with the number of groups. If |A|=100, is this approach even feasible? There is an assumption here that the partitions the available samples, but in practice you can have a sample be the intersection of two protected attributes, i.e., race and gender. In practice, poor performance often persists for the groups that happen to be intersection of protected attributes.
>
> - **Scalability with subgroup numbers.**
> Thanks for your remarks. In the Amazon-review experiment, $|A|=200$: we have 200 clients (subgroups). We further evaluated our method for different clients numbers with $|A|=50, 100, 300, 400$, as shown in appendix A.2. The results indicate that our method is consistently superior in different group numbers. Simultaneously this may result in high-memory complexity, because we need to train and save $|A|$ subgroup predictors in the lower-level optimization. To address this, we could randomly sample a subset of subgroup $|A^{’}|<|A|$ at each training epoch and solve the bi-level objective.
>
> - **Single and multiple protected attributes.**
> We are currently focusing on a single protected attribute such as gender. The algorithm could be practically extended to multi-protected attributes. Consider gender and race, then we can create subgroups solely on gender and race (i.e., group 1 = male, group 2 = female, group 3 = black, group 4 = white, with some overlaps between the subgroups).  Then we could run our framework by learning four subgroup-specific predictors and then learning a fair-informative predictor at upper-level. Specifically, samples with multi-attribute (such as gender, race) are optimized within different subgroups. Since our algorithm enforces all identical subgroup optimal predictors, it could achieve fair results across gender and race. Besides, this is merely an empirical extension; the exact theoretical assumptions for the provable fairness remain opening.
>
> > 3.Link From Theorems to Bilevel Formulation: It is not clear to me how the propositions and theorems in the paper suggest a bilevel formulation. It is still possible to relax the bilevel problem into a single objective with several penalties for each group. Not sure what in the theorems really necessitates a bilevel formulation.
>
> Our practice is motivated by Corollary 4.1, a fair and informative predictor should be closed to all the subgroup-specific optimal predictors, which motivates us to (1) learn subgroup optimal predictors, (2) then learn a *global* fair predictor to be close to all subgroup predictors.
>
> In practice, if we only have limited samples per subgroup (e.g., Amazon reviews), we should use the fair predictor as a prior for facilitating learning the subgroup specific predictors (lower-level). Theorem 4.1 further proves that if $Q$ is informative, the generalization error of subgroup specific predictor could be controlled.

---

> > ### Author Response · Authors · 2022-08-02
> > **Response 2/2**
> >
> > > 4.Take away from the theorems: Several of the theorems are in terms of the bayes optimal classifier for either a group or all data. Since the bayes classifier is usually infeasible to compute in practice, it is a bit hard to translate these insights to practice.
> >
> > We agree with your viewpoint. Indeed, because the ground-truth distribution is unknown, the A-Bayes optimal classifier $E[Y|X,A]$ cannot be exactly computed in practice.  As a result, we proposed several analyses such as (1) how to approximate A-Bayes predictor from a specific hypothesis family (Corollary 4.1), (2) how to approximate A-Bayes predictor from the data (Theorem 4.1).
> >
> > > 5.Challenge with different structural groups: I wonder if the authors can speak to issues that occur when, say two groups just have different structural properties. Consider a 1-d case where a group of points have two regression parameters with positive slope, but the other groups has true parameters with negative slope. I am not sure how the formulation will handle that. Perhaps this is a limitation of the current setup?
> >
> > Indeed our framework could not handle this extreme scenario, because we assume similar A-Bayes predictors $E[Y|X,A]$ across subgroup $A$. In this case, $E[Y|X, A=0]=  X$ and $E[Y|X, A=1]=-X$.  Then learning one informative predictor from these two groups is impossible. Specifically, the output $Y$ is completely different w.r.t. subgroup $A$ when given the same input $X$.  Then we were unable to develop an informative model that could simultaneously control the error of these two subgroups. We have added a limitation part to discuss this in Appendix A.1.
> >
> > > 6.Missing Literature on Bilevel Optimization: There is an emerging rich literature on bilevel optimization that this paper did not reference or discuss. Once can often solve these problems with implicit differentiation or unrolling(as is done here). A discussion of these points is needed here.
> >
> > Thank you for your advice! We now add related works on bilevel optimization in appendix A1. Due to the page limit, we will finally update it in the next version of our manuscript.

---

> > > ### Comment · Reviewer_wL7w · 2022-08-08
> > > **Response addresses my concerns**
> > >
> > > Thanks for the clarifications, the response addresses the concerns that I had. I see the method can actually scale up to group sizes of 400, which is already surprising to me. Like the authors mention, I still there is a trade-off here, it is unclear to me where the limit should be, i.e., would one want to keep training a predictor per group? At what point would should a proposal become untenable. That said, perhaps future work can explore these ideas. Indeed, the discussion of literature in bilevel optimization improves the work.  I'll be keeping my score.

---

> > > > ### Author Response · Authors · 2022-08-08
> > > > **Thanks for your feedback**
> > > >
> > > > Thanks for your feedback! When facing many subgroups, we think there exists a tradeoff between the memory complexity and training stability.  We will include this point in the limitation/discussion.

---

### Author Response · Authors · 2022-08-02
**Paper Update**

Dear all,

Thanks for the comments and suggestions! Due to page constraints, we included additional experimental results and discussions in Appendix A (in the updated main paper). We hope the responses could clarify your confusions, and we are quite happy to provide additional explanations.

---

### Meta-Review · Area_Chair_RnTa · 2022-08-26

**Recommendation:** Accept
**Confidence:** Certain

**Metareview:**

All reviewers acknowledged the contribution of the paper and its focus on the multiple (potentially a large number of) subgroup’s sufficiency gap. This is also a practically challenging setting, with each subgroup having only a very limited number of samples. The paper presents a bi-level optimization solution that shows favorable performance and is believed to be a solid and novel contribution to the relevant literature. The reviewers are unanimously happy with the reported results and the one added during the rebuttal.


**Award:**

No

---

### Decision · Program_Chairs · 2022-09-14

Accept